# Using portable low-resolution spectrometers to evaluate TCCON biases in North America

Nasrin Mostafavi Pak [1,2,3], Jacob K. Hedelius [1,a], Sébastien Roche [1,b], Liz Cunningham [1], Bianca Baier [5,7], Colm Sweeney [5], Coleen Roehl [4], Joshua Laughner [6], Geoffrey Toon [6], Paul Wennberg [4], Harrison Parker [4], Colin Arrowsmith [1], Joseph Mendonca [1,2], Pierre Fogal [1], Tyler Wizenberg [1], Beatriz Herrera [1,3], Kimberly Strong [1], Kaley A. Walker [1], Felix Vogel [2], and Debra Wunch [1]

[1]Department of Physics, University of Toronto, Toronto, Ontario, Canada
[2]Environment and Climate Change Canada, Climate Research Division, Toronto, Ontario, Canada
[3]Department of Physical and Environmental Sciences, University of Toronto Scarborough, Toronto, Ontario, Canada
[4]Division of Geological and Planetary Sciences and Division of Engineering and Applied Science, California Institute of Technology, Pasadena, CA, USA
[5]NOAA Global Monitoring Laboratory, Boulder, CO, USA
[6]Jet Propulsion Laboratory, California Institute of Technology, Pasadena, CA, USA
[a]Now at Space Dynamics Laboratory, Utah State University, UT, USA
[b]Now at John A. Paulson School of Engineering and Applied Sciences, Harvard University, Cambridge, MA, USA
[7]Cooperative Institute for Research in Environmental Sciences, University of Colorado-Boulder, Boulder, CO, USA

**Correspondence:** Nasrin Mostafavi Pak (nasrin.mostafavipak@mail.utoronto.ca)

**Abstract.** EM27/SUNs are portable solar-viewing Fourier Transform Spectrometers (FTSs) that are being widely used to constrain measurements of greenhouse gas emissions and validate satellite trace gas measurements. On a six-week-long campaign in the summer of 2018, four EM27/SUNs were taken to five Total Carbon Column Observing Network (TCCON) stations in North America to measure side-by-side to better understand their durability, as well as the accuracy and precision of retrievals from their trace gas measurements and to constrain site-to-site bias among TCCON sites. We developed new EM27/SUN data products using both previous and current versions of the retrieval algorithm (GGG2014 and GGG2020) and used coincident AirCore measurements to tie the gas retrievals to the World Meteorological Organization (WMO) trace gas standard scales. We also derived airmass-dependent correction factors for the EM27/SUNs. Pairs of column-averaged dry-air mole fractions (denoted with an X) measured by the EM27/SUNs remained consistent compared to each other during the entire campaign, with a 10-minute averaged precision of 0.3 ppm for $XCO_2$, 1.7 ppb for $XCH_4$ and 2.5 ppb for XCO. The maximum biases between TCCON stations were reduced in GGG2020 relative to GGG2014 from 1.3 ppm to 0.5 ppm for $XCO_2$ and from 5.4 ppb to 4.3 for $XCH_4$ but increased for XCO from 2.2 to 6.1 ppb. The increased XCO biases in GGG2020 are driven by measurements at sites influenced by urban emissions (Caltech and AFRC) where the priors overestimate surface CO. In addition in 2020, one EM27/SUN instrument was sent to the Canadian Arctic TCCON station at Eureka and side-by-side measurements were performed in March–July. In contrast to the other TCCON stations that showed an improvement in the biases with the newer version of GGG, the biases between Eureka's TCCON measurements and those from the EM27/SUN degraded with GGG2020, but this degradation was found to be caused by a temperature dependence in the EM27/SUN oxygen retrievals that is not apparent in the GGG2014 retrievals.

# 1  Introduction

Our knowledge of the global carbon cycle has considerably improved in recent years, with the development of space-based and ground-based remote sensing techniques that produce measurements of column-averaged dry-air mole fractions of greenhouse gases (GHGs) and other trace gases in the atmosphere (Jacob et al., 2016; Hakkarainen et al., 2016; Liu et al., 2017; Crowell et al., 2019; Qu et al., 2021). Remote sensing measurements from space are valuable for their global spatial coverage that enable GHG levels to be measured in regions of the world that are not easily accessible for ground-based measurements. To better constrain global GHG emissions and reduce bias in space-based measurements, ground-based remote sensing instruments are used for validation of satellite retrievals (Yoshida et al., 2013; Wunch et al., 2017; Sha et al., 2021a).

The Total Carbon Column Observing Network (TCCON) (Wunch et al., 2011) serves an important role in the validation of space-based instruments such as the Orbiting Carbon Observatories (OCO-2 and OCO-3) (Crisp, 2015; Eldering et al., 2019), the Greenhouse Gases Observing Satellites (GOSAT and GOSAT-2) (Yokota et al., 2009; Suto et al., 2021), TanSat (Liu et al., 2018), and the TROPOspheric Monitoring Instrument (TROPOMI) (Veefkind et al., 2012). TCCON consists of twenty-eight high-resolution Fourier Transform Spectrometers (FTSs), located on four continents, that record solar absorption spectra. Column averaged dry-air mole fraction (Xgas) of GHGs such as $CO_2$, $CH_4$, $N_2O$ and $H_2O$ as well as other trace gases such as CO and HF are retrieved from the recorded spectra (Wunch et al., 2011).

The GGG software (developed at the Jet Propulsion Laboratory) is used to retrieve Xgas values from TCCON spectra. GGG contains several programs, including one to convert interferograms into spectra (I2S), and GFIT, a nonlinear least squares spectral fitting algorithm that iteratively scales an a priori profile to generate an absorption spectrum that best matches the measured spectrum. GFIT retrieves total column amounts of the trace gases of interest from which the Xgas values are computed. In 2021, there was a transition from the previous version of the retrieval algorithm, GGG2014 (Wunch et al., 2015) to the latest version GGG2020 (Laughner et al., 2020), and the TCCON retrievals using GGG2020 were publicly released in April 2022.

Since TCCON is used for satellite validation and carbon cycle scientific studies, minimizing retrieval errors is crucial. The error budget of TCCON for Xgas was assessed by performing a sensitivity test on sources of uncertainty in GGG. The study of Wunch et al. (2015) suggests uncertainties below 0.25% (1 ppm) for $XCO_2$, 0.5% (9 ppb) for $XCH_4$ and 4% (4 ppb) for XCO based on the retrievals using GGG2014. Using GGG2020, the error budget improves to 0.16% (0.64 ppm) for $XCO_2$ , 0.34% (6.1 ppb) for $XCH_4$ and 1.4% (1.4 ppb) for XCO.

There are certain practices in place to ensure site-to-site consistency between TCCON observations. First and foremost, each TCCON station is equipped with nearly identical spectrometer hardware, and each dataset is analyzed using a consistent version of the GGG software, including identical spectroscopy. Other practices include regular measurements of the optical alignment of the instruments, quantified by their Instrumental Line Shape (ILS), calibration of surface pressure measurements, and comparisons with airborne measurements to scale gas retrievals to the World Meteorological Organization (WMO) trace gas standard (Wunch et al., 2010, 2015; Messerschmidt et al., 2011; Geibel et al., 2012).

TCCON spectrometers (Bruker IFS125HR) are large and difficult to relocate, making regular side-by-side comparisons between TCCON stations essentially impossible. Portable FTS instruments, on the other hand, can be deployed at different

locations. The EM27/SUNs (by Bruker Optics GmbH) are portable solar-viewing FTS instruments with a lower spectral resolution (0.5 cm$^{-1}$) than TCCON (0.02 cm$^{-1}$) that can be used to measure total column abundances of $CO_2$, $CH_4$, $H_2O$, and CO in the atmosphere in the same spectral region as the TCCON measurements (Gisi et al., 2012; Frey et al., 2015).

EM27/SUNs can be employed as a common reference or "standard" instrument identifying potential biases between TCCON stations by collecting coincident measurements performed at different times of the day and on multiple days at each station. EM27/SUNs are straightforward to operate, can be controlled remotely after set-up, and are easily shipped. Therefore, there is potential for using EM27/SUNs as a traveling "standard" between multiple TCCON sites to perform TCCON measurement intercomparisons. In addition to comparisons with TCCON instruments, EM27/SUNs are used independently for satellite validation in regions where there are no TCCON stations (Sha et al., 2021b; Frey et al., 2021; Jacobs et al., 2020). In this case, it is crucial to ensure that EM27/SUNs used within this validation network are on the same scale as TCCON in accordance with WMO standards and retrievals from their measurements are stable over time, particularly after shipping.

Hedelius et al. (2017) have assessed the biases in $XCO_2$ and $XCH_4$ between four TCCON sites in the United States by performing side-by-side measurements with two EM27/SUNs used as traveling standards and found an average site-to-site bias between the TCCON retrievals of 0.3 ppm in $XCO_2$ and 3 ppb in $XCH_4$. Hedelius et al. made several suggestions to improve and extend their results that we implemented in this study. They suggested repeating the campaign and adding XCO comparisons to the analysis. They also suggested performing coincident profile and ground-based measurements to better characterize EM27/SUN biases. Lastly, they suggested that using more than two EM27/SUNs would better identify if measurements from an instrument drift during the campaign. In addition to the four TCCON sites visited by Hedelius et al. (2017), we added two additional TCCON sites in Canada: one in East Trout Lake, Saskatchewan, and the other one at Eureka, Nunavut. For logistical reasons we only shipped one EM27/SUN to Eureka, but had 3–4 EM27/SUNs in all other locations. Furthermore, in this study, we focus on quantifying improvements in the retrieval algorithm by comparing old and new versions (GGG2014 and GGG2020). We also derive EM27/SUN-specific corrections to the retrievals for each version of GGG.

This paper is organized as follows: In Sect. 2, we briefly describe the instrumentation and measurement procedures used in the campaign and provide details about the measurements performed at each TCCON site. In Sect. 3, we describe the data processing procedures and details of the post-retrieval corrections. In Sect. 4, we present the biases between EM27/SUN and TCCON retrievals at each site, and in Sect. 5 we discuss the improvements achieved and provide suggestions for future work.

## 2    Campaign description

EM27/SUNs are portable solar-viewing FTSs with a spectral resolution of 0.5 cm$^{-1}$ and are described in detail in Gisi et al. (2012). The built-in sun tracker, along with the CamTracker software, tracks the sun during the day (Gisi et al., 2012). The original design includes one InGaAs detector, and records in the spectral range of 5500–11000 cm$^{-1}$, measuring absorption features from $CO_2$, $CH_4$, $H_2O$, and $O_2$. Retrievals of XCO are made possible by adding a second detector (extended InGaAs) and a long-pass filter to measure the 4000–5500 cm$^{-1}$ spectral region (Hase et al., 2016).

The FTS instruments used at TCCON sites are IFS125HR spectrometers (by Bruker Optics GmbH) operated at a spectral resolution of 0.02 cm$^{-1}$. The TCCON extended InGaAs detectors are sensitive between 3800–11000 cm$^{-1}$, similar to the total range covered by the two EM27/SUNs detectors (Wunch et al., 2011). Each TCCON FTS is coupled to a solar tracker.

In this campaign, we perform side-by-side measurements with the extended-range EM27/SUNs and all TCCON instruments in North America. Here, we describe the TCCON sites and provide details about the measurement procedures.

## 2.1 Sites Descriptions and Measurement Timeline

Locations of the TCCON sites we visited during the summer 2018 campaign, the spring-summer 2020 campaign, and the base station in Toronto are presented in Figure 1. Table 1 summarises details of the campaigns. The summer 2018 campaign started with four EM27/SUNs, with two-letter instrument IDs of "ta", "tb", "tc", and "dn", visiting the Caltech TCCON station (denoted "ci"), the Armstrong Flight Research Center (AFRC) TCCON station ("df"), and the Lamont TCCON station ("oc"). Before moving on to the Park Falls TCCON station ("pa"), one of the EM27/SUN instruments ("dn") was shipped back to its original location to participate in other activities. We continued the campaign with three EM27/SUNs at Park Falls and the East Trout Lake TCCON station ("et"). At Park Falls, one of the EM27/SUN instruments ("ta") had a tracker failure, so we only measured with the other two EM27/SUN instruments. The problem was fixed at East Trout Lake and we continued measuring with three EM27/SUN instruments. One EM27/SUN ("tb") was shipped to the Eureka TCCON station ("eu") in February 2020 and the instrument performed side-by-side measurements with the Eureka TCCON instrument until July 2020.

During the road trip, we performed measurements using multiple EM27/SUNs placed in open air (i.e., outside) within 100 meters of the TCCON FTS. The TCCON instruments are housed inside a building or a container with a solar tracker on its roof, and the solar beam is directed into the instrument using multiple mirrors. A high accuracy pressure sensor accompanied the EM27/SUNs to account for any differences in pressure due to the slight difference in the height of the instrument set up. TCCON instruments run automatically during cloud-free times of the day when there is sufficient sunlight. We set up the EM27/SUNs in proximity to the TCCON sites to collect coincident measurements during daylight hours. During cloudy and rainy hours, the EM27/SUNs are stored indoors. In the following sections, we briefly describe the measurement sites and the measurement conditions during the campaign.

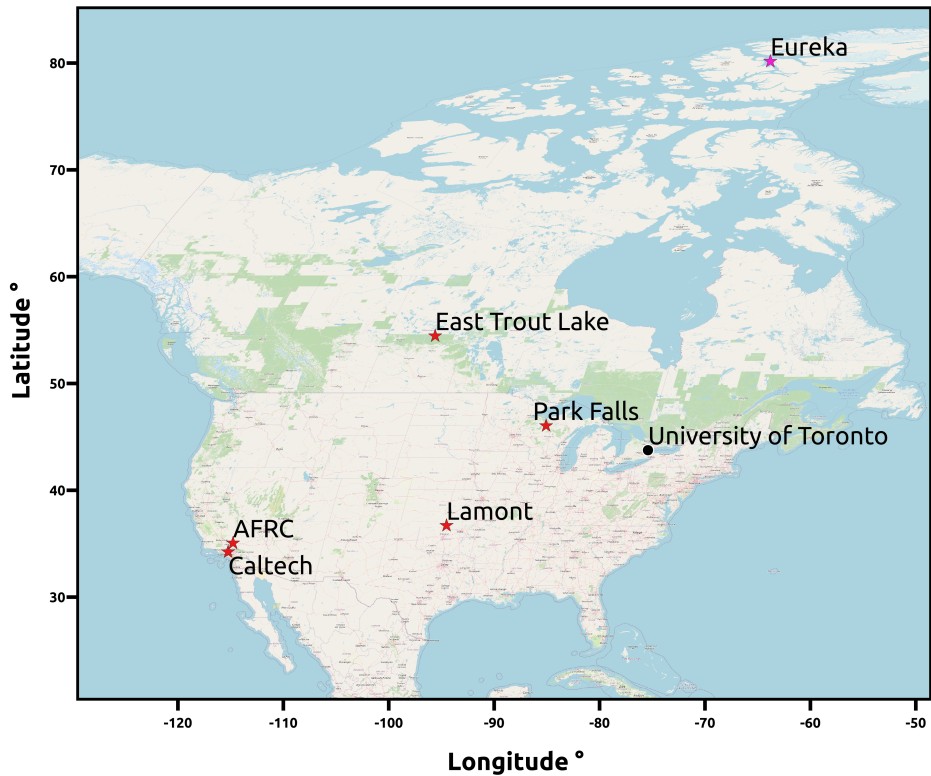

**Figure 1.** TCCON sites visited during the summer 2018 campaign are indicated by red stars located at Caltech ("ci"), AFRC ("df"), Lamont ("oc"), Park Falls ("oc"), and East Trout Lake ("et"). The magenta star indicates the Arctic site at Eureka ("eu") visited in spring and summer of 2020. Long term EM27/SUN measurements at the University of Toronto (black circle) have been used for post-retrieval corrections. The map is generated using OpenStreetMap (OSM) with Eckert III projection © OpenStreetMap contributors 2017. Distributed under the Open Data Commons Open Database License (ODbL) v1.0. (OpenStreetMap contributors, 2017).

### 2.1.1 Toronto - base station

Two of the EM27/SUN instruments used in this campaign belong to the University of Toronto ("ta" and "tb") and began measurements in Toronto in June 2017. We include only measurements after June 2018 in this analysis, as the instruments' optical benches were realigned in May, 2018. There is no TCCON station in Toronto but the data collected in Toronto was used for post-retrieval corrections. Measurements are performed on the 15th floor balcony of the McLennan Physical Laboratories at the University of Toronto (43.661°N, 79.399°W, 152 masl) on June 8–19 2018. Side-by-side measurements with the third

instrument ("tc") from Environment and Climate Change Canada (ECCC) were performed on June 20–21. All three instruments were subsequently shipped to Caltech.

**Table 1.** This table lists the TCCON site locations and the dates the EM27/SUNs were on-site, the number of days with successful measurements, the average number of spectra collected by each EM27/SUN, and the total number of spectra collected by the TCCON instrument during the visit. The ILS column indicates whether EM27/SUN ILS measurements were collected at that location. The number of Air-Core launches performed near the TCCON station during the dates listed is included in the final column. *instrument in brackets was not operational

| TCCON Site | Latitude °N | Longitude °W | Elevation (masl) | Dates | Days | EM27/SUNs* | Spectra count EM27/SUN | Spectra count TCCON | ILS | AirCore Launches |
|---|---|---|---|---|---|---|---|---|---|---|
| Caltech (ci) | 34.136 | 118.127 | 237 | 2018-07-06 – 2018-07-12 | 7 | ta,tb,tc,dn | 21,356 | 1,268 | Yes | - |
| AFRC (df) | 34.960 | 117.881 | 699 | 2018-07-13 – 2018-07-19 | 7 | ta,tb,tc,dn | 22,667 | 2,177 | No | 6 |
| Lamont (oc) | 36.605 | 97.486 | 320 | 2018-07-21 – 2018-07-29 | 5 | ta,tb,tc,dn | 17,744 | 921 | Yes | 9 |
| Park Falls (pa) | 45.945 | 90.273 | 442 | 2018-07-31 – 2018-08-07 | 4 | (ta),tb,tc | 4,436 | 406 | No | 4 |
| East Trout Lake (et) | 54.354 | 104.987 | 517 | 2018-08-09 – 2018-08-18 | 6 | ta,tb,tc | 14,910 | 770 | Yes | - |
| Eureka (eu) | 80.053 | 86.417 | 610 | 2020-03-04 – 2020-08-31 | 61 | tb | 131,713 | 5,166 | No | - |

### 2.1.2 Caltech

The campaign began at the California Institute of Technology (Caltech) TCCON site, which has been operational since 2012. Caltech is located in Pasadena, California within the South Coast Air Basin (SoCAB) that includes the city of Los Angeles, and is influenced by large urban emissions. The three EM27/SUNs from Toronto, as well as the EM27/SUN owned by Jet Propulsion Laboratory (JPL) and operated at Caltech ("dn") measured side-by-side from July 6 to July 12 on the roof of the Linde Laboratory for Global Environmental Science where the TCCON sun-tracker mirrors were located. The weather conditions at the time were mostly sunny with rare cases of passing clouds. We performed side-by-side measurements for all seven days on site.

### 2.1.3 AFRC

The TCCON site at the Armstrong Flight Research Center (AFRC) on Edwards Air Force Base, California, has been operating since July 2013. It is located 150 km northeast of Los Angeles in the western Mojave Desert (NASA, 2021). Since AFRC is located outside the SoCAB, measurements are not as strongly influenced by urban emissions. Measurements were performed with the four EM27/SUNs from July 13 to July 19. From July 16 to July 18, one of the EM27/SUNs was taken to a remote site collocated with the balloon launch location. The instruments at the TCCON site were placed on the ground next to the container within a few meters of the TCCON instrument. The weather conditions were mostly sunny and we performed measurements for seven days.

### 2.1.4 Lamont

The Lamont TCCON station is located at the Southern Great Plains Atmospheric Radiation Measurement site near Lamont, Oklahoma. Measurements were performed with the four instruments from July 21 to July 28. All the instruments were placed on a platform that was roughly 100 meters away from the TCCON instrument. The weather for most of the days was mostly

sunny with some passing clouds. July 26 and 28 were rainy or cloudy most of the day. Therefore we collected measurements for five total days.

### 2.1.5 Park Falls

The Park Falls TCCON station is located within the boreal forest in northern Wisconsin, co-located with the WLEF-TV 472 masl tall tower. The EM27/SUNs were set up outside the TCCON container on the ground. The weather conditions at Park Falls during our visit were mostly cloudy and rainy, limiting our measurements to four sunny days between July 31st to August 6th.

### 2.1.6 East Trout Lake

The East Trout Lake TCCON station is located within the boreal forest in Saskatchewan, Canada. The EM27/SUNs were set up on the rooftop of the laboratory building, next to the TCCON instrument's solar tracker. During our visit, forest fire plumes transported from British Columbia caused severe smoky conditions. Therefore, in addition to the cloudy days, thick smoke blocked the sunlight for many hours even under cloudless conditions. We performed six days of successful side-by-side measurements between August 10 to August 18.

### 2.1.7 Eureka

The Eureka TCCON station is located near Eureka, Nunavut, at the Polar Environment Atmospheric Research Laboratory (PEARL) Ridge Laboratory (Fogal et al., 2013; Barret et al., 2002). One of the EM27/SUNs ("tb") was sent to Eureka in February 2020 to perform side-by-side measurements with the TCCON instrument. The EM27/SUN was set up inside the lab just beside the 125HR instrument. The EM27/SUN sun-tracker was removed and a pick-off mirror was used to redirect some
155 of the parallel beam from the TCCON sun-tracker to the EM27/SUN. Side-by-side measurements were performed from March 2020 to July 2020 when the internal laser of the 125HR failed and TCCON measurements were stopped, summing to a total of 61 days.

## 2.2 Pressure measurements

Accurate surface pressure measurements are important for the retrieval algorithm to calculate the total column of dry air;
biases in surface pressure lead to biases in Xgas values. Therefore, it is necessary to ensure that the pressure measurements are accurate by calibrating the local TCCON pressure sensors; in this work, this was done using a single, portable pressure standard with high accuracy. We used a Digiquartz sensor with an accuracy of 0.08 hPa (Paroscientific Inc., 2011). At five of the TCCON sites, surface pressure is measured using a Setra pressure transducer (Coastal Environmental Systems, Inc. Barometer 270) with an accuracy of 0.3 hPa (Coastal Environmental Systems Inc.). At East Trout Lake, a GE-8100 pressure
sensor is used with an accuracy of 0.01% ($\approx$ 0.1 hPa ) (General Electric Company, 2012). In addition, during the road trip,

EM27/SUNs were accompanied by a Vaisala Weather Station (Model WXT536) that has a pressure sensor with accuracy of 0.5 hPa (Vaisala, 2017) as a back up.

For the EM27/SUN retrievals, we use the pressure measurements recorded by the local TCCON weather stations which have previously been calibrated against a pressure standard, with the exception of Park Falls where we used the Vaisala WXT536
pressure data for both TCCON and EM27/SUNs, since the pressure measurements made by the TCCON pressure sensor were not stable during the campaign. In addition, at AFRC and Lamont we applied additional corrections to the EM27/SUN pressures as they were deployed at a slightly different altitude than the 125HR tracking mirror at the TCCON site. In these cases, we used a Digiquartz pressure standard that was measuring at the same altitude as the EM27/SUNs to calculate the difference in surface pressure and added an offset of +0.1 hPa at AFRC and +0.3 hPa at Lamont to the original pressure value.
For more details on pressure calibration refer to Appendix B.

## 2.3  ILS measurements

Instrumental Line Shape (ILS) is a measure of the optical alignment of the instrument and imperfections in this alignment can cause biases in the retrievals. The ILS of an FTS can be described by two parameters: Phase Error (PE) and Modulation Efficiency (ME). For high resolution FTS instruments (TCCON) ILS is typically reported as function of optical path difference
(OPD) and for the low resolution EM27/SUNs these are typically reported at the maximum optical path difference. ILS values are not implemented into the GGG2014 and GGG2020 retrieval algorithms and are only used to evaluate the instruments' alignment qualitatively. TCCON guidelines require that the modulation efficiency deviates less than 5% from 1.0, over 0 to 45 cm OPD. A modulation loss of 1% in the EM27/SUN causes a bias of 0.1% in $XCO_2$ and 0.15% in $XCH_4$ (Hedelius et al., 2016). We aim to ensure that the EM27/SUN ME variations remain less than 1%.
Because the EM27/SUNs were moved from one site to another, we evaluated their optical alignment by measuring the ILS of all the EM27/SUNs at three TCCON stations (Table 1). For EM27/SUNs, we use a method introduced by Frey et al. (2015) and further developed by Alberti et al. (2022), in which we collect spectra from an external lamp (Quartz Tungsten-Halogen Lamp, Thorlabs) in the laboratory and use LINEFIT (version 14.0) to derive the ILS parameters from $H_2O$ lines in the 7000 and 7400 cm $^{-1}$ spectral region (Frey et al., 2015). Our method differs slightly because we use three different distances between the
lamp and the EM27/SUN, and we compute the standard deviation of the calculated ME across the three distances to evaluate the variability in the calculated ME.

An ILS test was performed on the three Toronto instruments in May 2018, prior to shipment to Caltech. During the road trip, we performed ILS tests at three of the sites. At Caltech, ILS tests for all four EM27/SUNs were performed on the morning of July 10th. At Lamont, ILS tests were performed on July 21st and 22nd on all four instruments. At East Trout Lake, ILS tests
were conducted on August 11th and 12th on the three instruments. In addition, ILS characterizations were performed on three instruments in September 2018, after they returned to Toronto.

The results of ILS tests for the EM27/SUNs during the campaign are presented in the lower panel of Figure 2 as a function of time. The error bars represent the variability in calculated ME. The fluctuations in EM27/SUN modulation efficiency from site-to-site are less than 1% indicating that the instruments remained sufficiently well aligned.

The fluctuations in the measured ME values are larger than ones reported by Alberti et al. (2022) (less than 0.5% over a period of one month), but this could be due to different laboratory environments during the ILS measurements. Since the ILS method relies on water vapor lines, the room conditions such as relative humidity and temperature play an important role in the results and these parameters were not controlled in any of the rooms where the tests were performed, although they were measured. In addition, it was not possible to set up the lamp at the same distances from the EM27/SUN at each location which could lead to additional variations. Variations in the temperature and relative humidity in the room can cause errors in the ILS calculation. A sensitivity test showed a 10% bias in relative humidity measurements would lead to a 0.3% difference in retrieved ME. We used the Vaisala WXT536 weather station which hosts a humidity sensor that has an accuracy of 3%.

ILS tests are regularly performed at TCCON sites by collecting lamp spectra through an internal HCl cell (Hase et al., 2013) and using LINEFIT as the retrieval software (Hase et al., 1999). The results of ILS tests for the TCCON instruments are presented in the upper panel of Figure 2 as a function of optical path difference (OPD). ME at all the TCCON sites remains with the TCCON guidelines, and therefore we consider the TCCON instruments to be well aligned during the campaign.

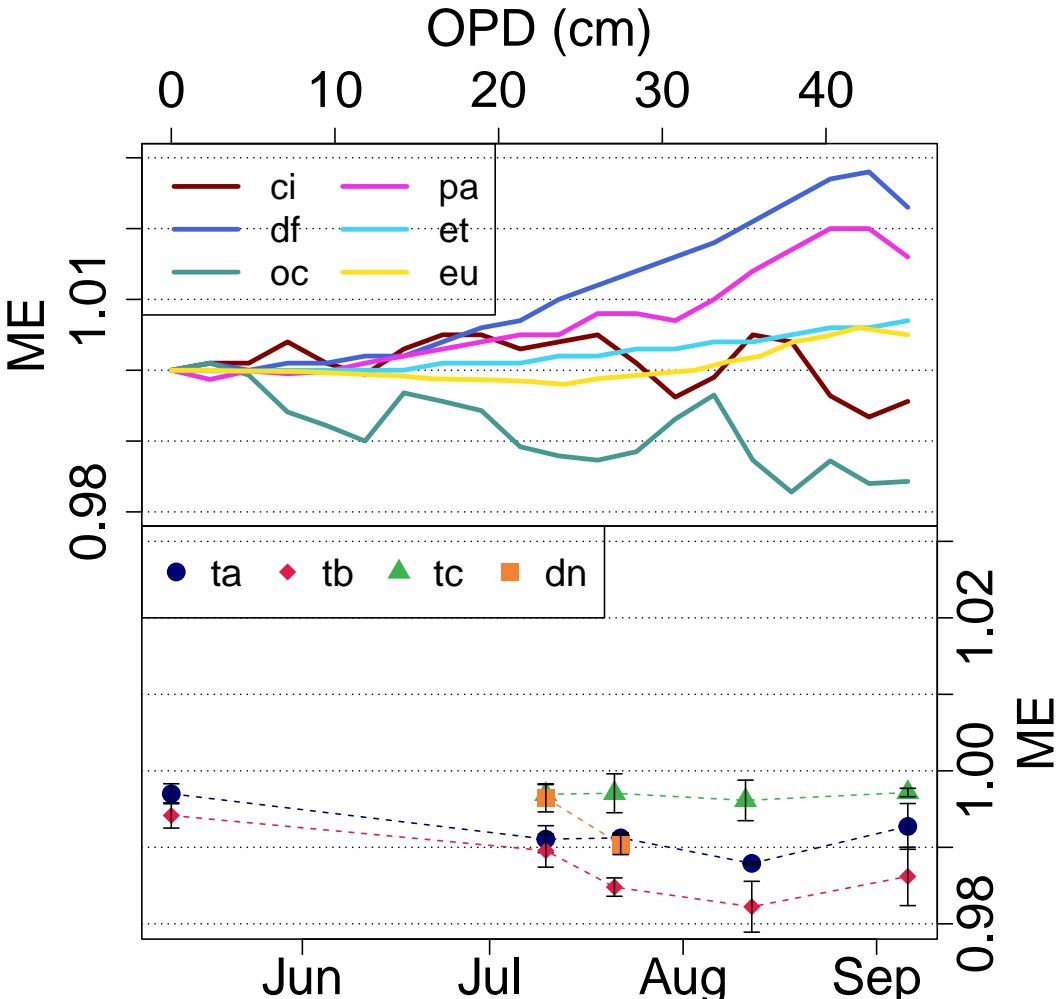

**Figure 2.** Modulation efficiency of TCCON instruments at different OPD (0-45 cm, x-axis top panel), and EM27/SUNs at maximum OPD (1.8 cm) over time (May 2018 to September 2018, x-axis bottom panel)

## 2.4 AirCore measurements

In order to achieve traceable accuracy, total column measurements from TCCON are tied to the WMO trace gas scale by comparing with calibrated airborne in situ measurement profiles that are simultaneously collected at the TCCON sites (Wunch et al., 2010, 2015; Messerschmidt et al., 2011; Geibel et al., 2012). One limitation with aircraft profiles is that they usually have an altitude ceiling of about 8-14 km (Sweeney et al., 2015). An alternative method for obtaining vertical profiles that extend higher is to use the AirCore sampling system (Tans, 2009; Karion et al., 2010; Tans, 2022; Baier et al., 2023). In this method, a coiled 100-meter long hollow tube with a small inner diameter of about 0.2–0.3 cm is launched using a balloon. The

AirCore is filled with a mixture of known trace gas mole fractions of interest prior to launch, and this gas evacuates during ascent. Upon descent, the nearly-empty AirCore fills with ambient air where the earliest sample is compressed into the topmost portion of the tube. Because molecular diffusion and Taylor dispersion acts slowly within this tubing coil (Tans, 2022), there is little mixing of the continuous air sample collected within the tube. The tube is then sealed upon landing, retrieved and quickly analyzed using a Picarro Cavity Ring-Down Spectrometer. AirCore altitude ceilings for balloon flights are typically set to 30 km asl, with trace gas profiles derived from approximately 27 km to the surface (Karion et al., 2010).

In collaboration with National Oceanic and Atmospheric Administration Global Monitoring Laboratory (NOAA GML), we used the AirCore system to obtain vertical profiles of $CO_2$, $CH_4$, and CO alongside concurrent EM27/SUN measurements at three of the measurement sites: AFRC, Lamont, and Park Falls. At AFRC, six AirCore launches were performed at a remote site, 30 km from the TCCON and EM27/SUN measurement site location due to airspace restrictions on July 16th, 17th and 18th. To facilitate nearby comparisons of AirCore trace gas data with the EM27/SUNs, one of the EM27/SUNs ("ta") was taken to the remote site, closer to where the AirCores landed (34.691 °N, 117.818 ° W, 810 masl), while the other three were performing side-by-side measurements with the TCCON spectrometer. At Lamont, nine AirCores were launched on July 23rd, 25th and 27th. At Park Falls, four AirCores were launched on July 31st and August 3rd. We did not successfully measure with the EM27/SUNs at the time of the launch on August 3rd due to clouds. A summary of dates, times and locations of the AirCore launches is presented in Table 2.

AirCore mole fractions registered to altitude levels are integrated following the method described in Wunch et al. (2010) to achieve an average total column mixing ratio that is comparable to the Xgas values retrieved from the EM27/SUN instruments. When integrating the AirCore column, we fill in the higher altitude mole fractions that were not measured by AirCore with the GGG a priori values and extrapolate the lowest altitude measurements to the surface. At the Lamont and Park Falls sites, in situ tower measurement data are available (Biraud et al., 2001; Andrews et al., 2014), and we use those measurements to extrapolate mole fractions to the surface. Figure 3 shows sample AirCore vertical profiles of $CO_2$ (in black) collected at the Lamont site on July 25th, 2018.

To compare column-integrated AirCore in situ profiles we perform retrievals on the EM27/SUN spectra using the AirCore profile as the a priori profile. We use the standard GGG a priori profile above the AirCore ceiling. This allows us to identify the spectroscopic scaling required to place the EM27/SUN measurements to the WMO scale.

**Table 2.** Summary of AirCore launches coincident with EM27/SUN measurements during the 2018 road trip.

| Site | Latitude °N | Longitude °W | Elevation (masl) | Date | Launch time (UTC) | Number of launches | Collocated EM27/SUN(s) |
|---|---|---|---|---|---|---|---|
| AFRC remote site | 34.691 | 117.818 | 810 | 2018-07-16 | 21:30 | 2 | ta |
| | | | | 2018-07-17 | 14:00 | 1 | ta |
| | | | | 2018-07-17 | 21:30 | 1 | ta |
| | | | | 2018-07-18 | 18:00 | 2 | ta |
| Lamont | 36.605 | 97.486 | 320 | 2018-07-23 | 17:00 | 2 | ta,tb,tc,dn |
| | | | | 2018-07-25 | 17:00 | 4 | ta,tb,tc,dn |
| | | | | 2018-07-27 | 17:00 | 3 | ta,tb,tc,dn |
| Park Falls | 45.945 | 90.273 | 442 | 2018-07-31 | 17:00 | 2 | tb,tc |
| | | | | 2018-08-03 | 17:00 | 2 | – |

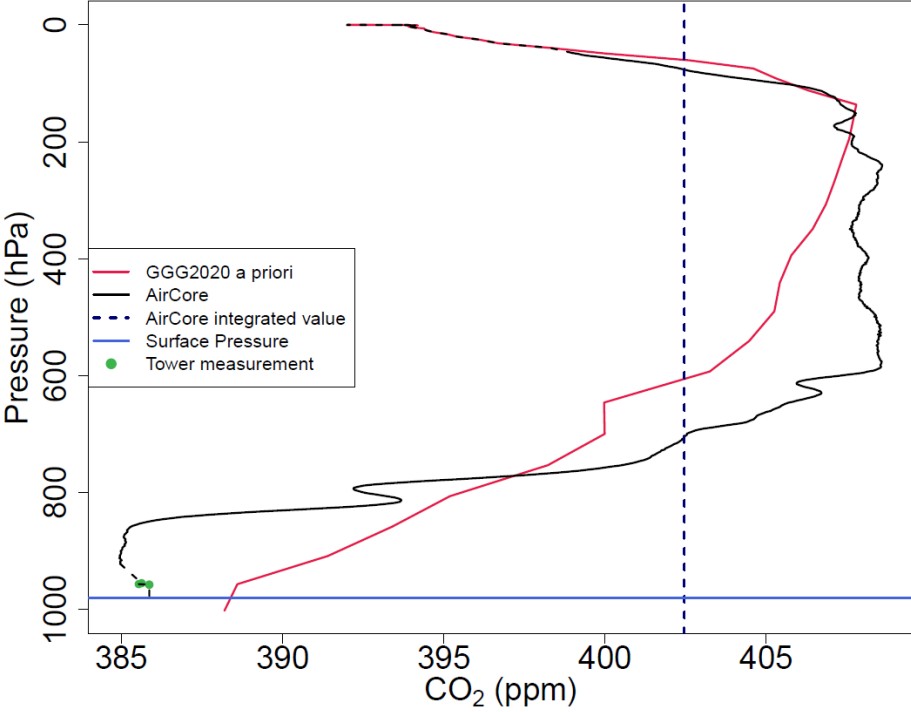

**Figure 3.** Sample of AirCore measurement (black line), a priori profile used by GGG2020 (red line), in situ tower measurements (green dots) and surface pressure measured by the ground based sensor (blue horizontal line) at Lamont on July 25, 2018. The dashed black line shows the portion of the AirCore profile that was extrapolated using the GGG2020 a priori profile above the AirCore ceiling altitude, or interpolated using the tower measurements below the profile. The navy blue dashed vertical line depicts the average calculated column value using AirCore.

## 3 Data processing

In this study, we use two versions of the TCCON and EM27/SUN retrieval software: GGG2014 (Wunch et al., 2015), and GGG2020 (Laughner et al., 2020) to evaluate improvements in the retrieval and persistent biases between the instruments. Several updates have been made to the a priori profiles (described in Laughner et al., 2022), and changes to the spectroscopy include improvements in the spectroscopic linelist, and the addition of a non-Voigt lineshape model for $CO_2$, $CH_4$, and $O_2$ (Mendonca et al., 2016, 2017, 2019). A detector non-linearity correction has also been applied to the interferograms in GGG2020. To process EM27/SUN spectra, we use the EM27 GGG Interferogram processing suite (EGI) (Hedelius and Wennberg, 2023a, b). We use the same spectral regions and linelist to retrieve $O_2$, $XCO_2$, $XCH_4$, and XCO from both instrument types.

An a priori prediction of the atmosphere at the location and time of the measurement is required to model the atmospheric transmittance spectra. In GGG2014, a priori vertical profiles of pressure, temperature, and water vapour at the location of the measurements for each day are obtained from the National Centers for Environmental Prediction and National Center for Atmospheric Research (NCEP/NCAR) reanalyses (Kalnay et al., 1996). The GGG2014 retrieval grid is on 71 vertical equally spaced 1-km levels defined from the ground to 70 km, and one prior is produced representing local noon on each day. For GGG2020, 3-hourly vertical profiles are obtained from Goddard Earth Observing System- Forward Processing for Instrument Teams (GEOS-FPIT) atmospheric data assimilation system (Lucchesi, 2015) and the retrieval grid has 51 vertical levels with increased spacing with altitude (0.4 km at sea level to 2.4 km at 70 km) (Laughner et al., 2022).

After GFIT has completed a retrieval, several "post-processing" programs apply corrections to the retrieved total columns. First, column-averaged dry-air mole fractions (Xgas) are calculated using the ratio of the total column abundance of the gas of interest to that of oxygen, multiplied by the mole fraction of oxygen in the atmosphere. Second, an airmass-dependent correction is applied to the Xgas retrievals, and third, a scaling is applied to the Xgas retrievals to place them on the WMO trace-gas standard scale (Wunch et al., 2011).

Finally, the data are passed through a set of quality control (QC) tests, and QC flags are added to each individual measurement based on predefined criteria. The data with flag=0 have successfully passed all the QC filters and are used in further analyses. In addition to the usual filtering done by GGG post-processing, we also add an additional condition to filter out data with poor solar tracker pointing. This additional tracking filter is described in Appendix A.

We derive EM27/SUN-specific correction factors to the data and compare those to the correction factors calculated for TCCON. In the next section, we describe the procedure to derive EM27/SUN-specific post-processing corrections.

### 3.1 EM27/SUN post-processing corrections

The precision of the measurements and the instrument-to-instrument agreement are important because the EM27/SUNs are anticipated to function as satellite validation tools, to investigate biases between TCCON stations, and to estimate large urban or point-source GHG emissions. In this study, we create a new EM27/SUN product with minimized instrument-to-instrument bias, minimized airmass-dependent artefacts, and an independent scaling to the WMO trace gas scale. In this section, we describe the methods used to apply additional corrections to the EM27/SUN retrieval products.

### 3.1.1 EM27/SUN airmass-dependence correction

Retrieved Xgas values have a small solar zenith angle (or airmass) dependence caused by spectroscopic inaccuracies and instrumental differences. We attempt to isolate and correct for the spectroscopic component of this artefact using an empirical correction described by Wunch et al. (2011). Improvements in the spectroscopic line shape model and the airmass dependent correction model within GGG2020 have reduced the airmass-dependent artefacts compared with GGG2014 (Mendonca et al., 2016, 2017, 2019).

In this model, an asymmetric function ($A(t_i)$) representing true variations in Xgas over the course of the day and a symmetric function ($S(\theta_i)$) representing the airmass dependent artefact are used to fit to the daily measured Xgas values during the course of the day (Wunch et al., 2011).

$$y_i = \hat{y}[1 + \alpha A(t_i) + \beta S(\theta_i)] \tag{1}$$

where $y_i$ is the Xgas from each spectrum and $\hat{y}$ is the mean value of Xgas on that day. $A(t_i)$ and $S(\theta_i)$ are defined as (Wunch et al., 2011):

$$A(t_i) = sin(2\pi(t_i - t_{noon})) \tag{2}$$

where $t_i$ and $t_{noon}$ are in unit of days.

$$S(\theta_i) = \left(\frac{\theta_i + \theta_0}{90 + \theta_0}\right)^p - \left(\frac{45 + \theta_0}{90 + \theta_0}\right)^p \tag{3}$$

where $\theta_i$ is in degrees and $\theta_0$ and $p$ are empirically found to be $13°$ and 3, respectively. $\alpha$ and $\beta$ are found by minimizing the difference between the measured $y_i$ and the fitted functions. Airmass dependent correction is then applied to Xgas by:

$$y_c = \frac{y_i}{[1 + \beta S(\theta_i)]} \tag{4}$$

where $\beta$ is the airmass dependent correction factor (ADCF) and $y_c$ is the airmass corrected Xgas value.

To derive the airmass-dependent correction factors (ADCFs) for the EM27/SUNs, we use the long term record of measurements in Toronto from 2018 to 2021 with four EM27/SUNs ("ta", "tb", "tc" and "td" [1]) to calculate an average ADCF value for each gas. Although Toronto is under the influence of urban emissions, our analysis showed that the enhancements due to traffic emissions are not symmetric around noon and therefore would not interfere with the airmass dependent calculations. The derived ADCFs are then applied to all further measurements to minimize any spectroscopically driven airmass-dependencies. Table 3 summarises ADCF values derived for GGG2014 and GGG2020. We choose an ADCF of 0 for XCO since CO has a large dynamic range compared to the other gases and symmetric components can exist in the diurnal CO patterns due to traffic. Airmass-dependent corrections are applied to $XCO_2$, $XCH_4$, and XCO in both GGG2014 and GGG2020, and in GGG2020, an airmass-dependent correction is also applied to a parameter called "Xluft", which is the column-averaged amount of dry air, formerly called Xair in GGG2014.

---

[1] "td" is an ECCC-owned EM27/SUN that started measurements in 2019. Its data are not used in the analysis of this paper except to derive the airmass-dependent correction factor.

To confirm that our newly derived ADCFs reduce the airmass dependence of the EM27/SUN measurements, we plot the daily anomaly of $XCO_2$ and $XCH_4$, calculated by taking the difference between 1-minute averages and the daily median, against the solar zenith angle at the time of the measurement. For GGG2014, the airmass-dependent correction factor calculated for $XCO_2$ was not significantly different than that calculated for TCCON, and therefore we use the same correction factor. For $XCH_4$, the airmass dependence of the EM27/SUN data is negligible, and applying the TCCON correction factors increases the airmass dependence. Therefore we choose an ADCF of 0. Figure 4 shows the $XCO_2$ and $XCH_4$ anomalies for the EM27/SUN with the longest measurement record in Toronto that also performed measurements at 4 TCCON stations in 2018 ("ta").

TCCON applies ADCFs for GGG2020 differently than in GGG2014. In the GGG2020 TCCON post-processing procedure, ADCF values are calculated for each retrieval window *before* averaging them (Laughner et al., 2020), whereas in GGG2014, ADCFs are calculated and applied for each gas after averaging different retrieval windows. Additionally, in GGG2020, $\theta_0$ and $p$ vary from window to window in order to best capture the airmass dependence, whereas in GGG2014 all gases used $\theta_0 = 13°$ and $p = 3$. For EM27/SUNs, we follow the same method as in GGG2014 and derive and apply ADCFs for each gas *after* averaging individual windows. It is therefore not possible to directly compare TCCON and EM27/SUN ADCF values for gases with multiple windows (i.e., $CO_2$ and $CH_4$); it is expected that they will be different. Figure 5, shows that using our derived ADCFs eliminates the EM27/SUN airmass-dependent artefacts for $XCO_2$ and $XCH_4$ effectively.

**Table 3.** Airmass-dependent Correction Factors (ADCF) used for TCCON and EM27/SUNs. A dash (–) indicates that the value is not calculated or applicable.

|         | TCCON GGG2014 ADCF | TCCON GGG2020 ADCF | EM27/SUN GGG2014 ADCF | EM27/SUN GGG2020 ADCF |
|---------|--------------------|--------------------|-----------------------|-----------------------|
| Xluft   | –                  | 0.00053            | –                     | 0.0027                |
| $XCO_2$ | -0.0068            | –                  | -0.0068               | -0.0049               |
| $XCH_4$ | 0.0053             | –                  | 0                     | -0.0045               |
| XCO     | -0.0483            | 0                  | 0                     | 0                     |

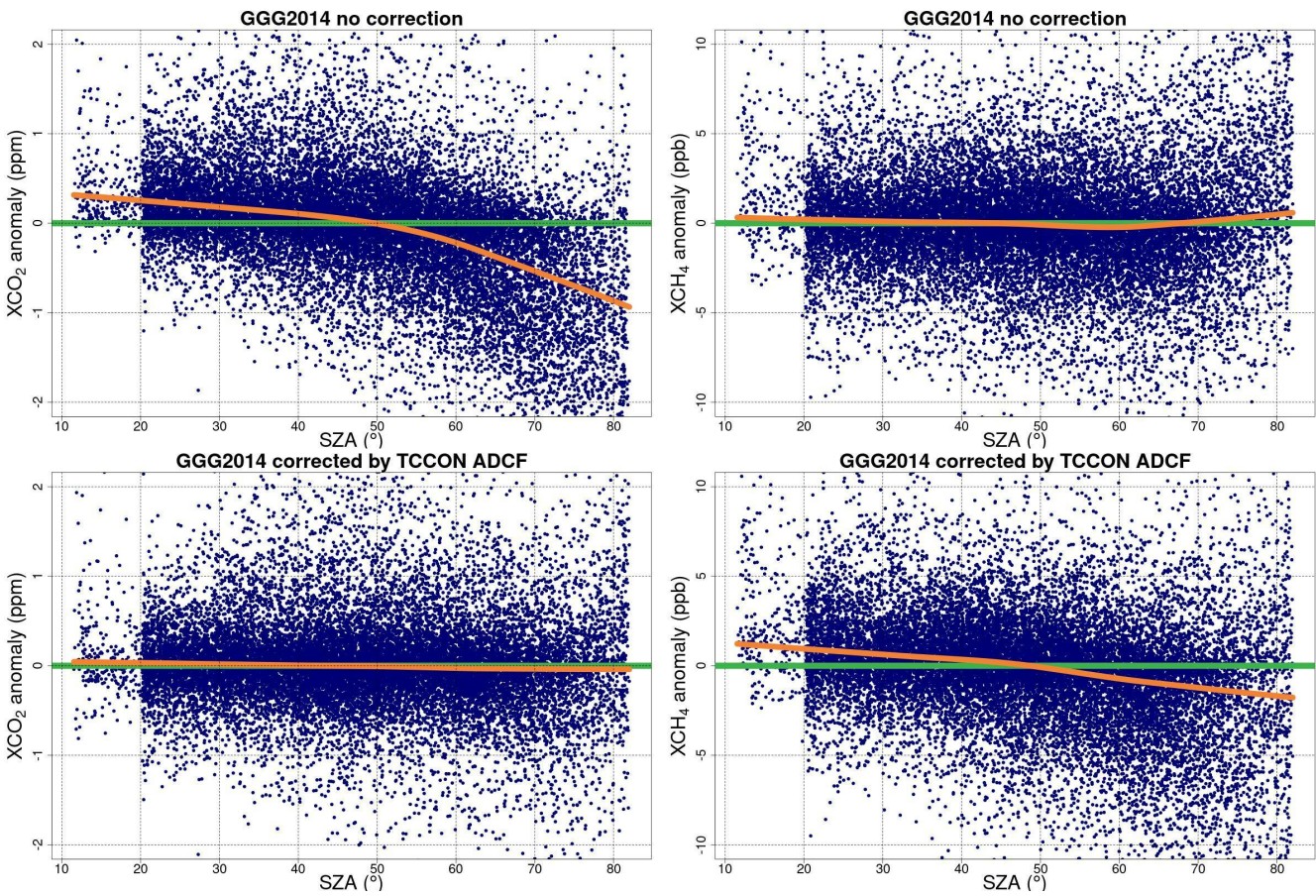

**Figure 4.** Daily $XCO_2$ and $XCH_4$ anomalies of the EM27/SUN ("ta") measured in Toronto from 2017 to 2021 and during the road trip in 2018 vs. solar zenith angle using GGG2014: top panel no ADCF applied and the bottom panel TCCON ADCF applied. The orange line is the Lowess curve fit. The green horizontal line is the y=0 line.

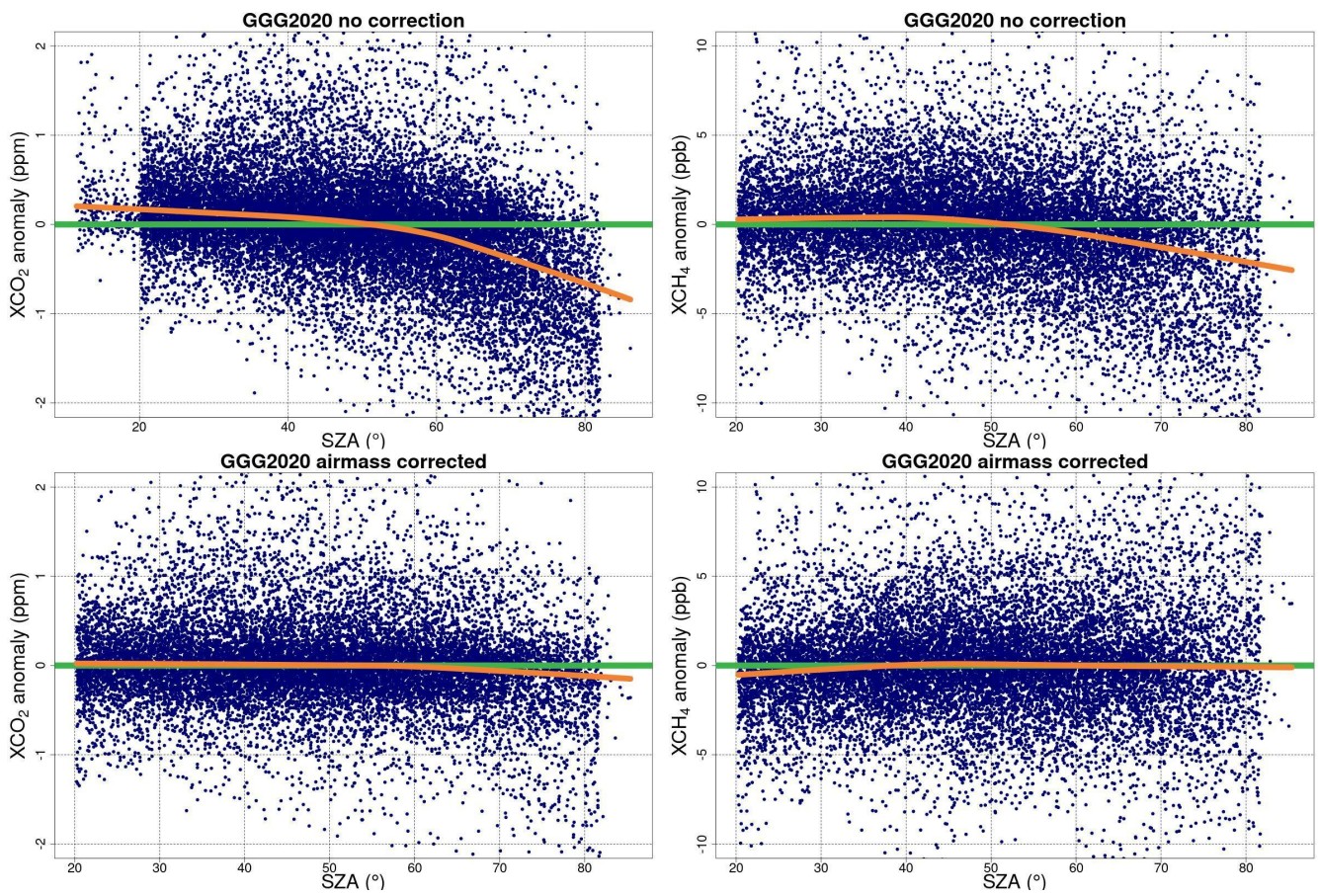

**Figure 5.** Daily XCO$_2$ and XCH$_4$ anomalies of the EM27/SUN ("ta") measured in Toronto from 2017 to 2021 and during the road trip in 2018 vs. solar zenith angle using GGG2020: top panel no ADCF applied, and the bottom panel the derived ADCF for the EM27/SUNs are applied. The orange line is the Lowess curve fit. The green horizontal line is the y=0 line.

### 3.1.2 EM27/SUN instrument-to-instrument bias correction

Small biases in Xgas are expected to exist between the EM27/SUNs due to the differences in the instrument alignment (i.e., ILS) (Alberti et al., 2022). As long as these biases remain constant in time, including after shipping, a simple additive correction

can place all the EM27/SUNs onto the same scale. Previous studies have found biases among retrievals from EM27/SUN instruments of up to about 0.8 ppm for XCO$_2$ and 5 ppb for XCH$_4$ (Chen et al., 2016; Hedelius et al., 2016; Frey et al., 2019a).

To correct for the instrument-to-instrument biases, we apply offsets such that all the instruments are on the same scale as the JPL (dn, serial number : 42) EM27/SUN for XCO$_2$ and XCH$_4$. We chose this instrument because most of the other EM27/SUNs in North America have visited the Caltech site and have performed side-by-side measurements there, allowing us

to place all North American EM27/SUNs on the same scale. The JPL EM27/SUN does not measure XCO, so we use "ta" to set the XCO offsets.

However, JPL's EM27/SUN ("dn") did not visit every TCCON station during the field campaign, so we first calculate EM27/SUN biases with respect to each other referencing "tb", which measured reliably at each site. We subtract coincident 10-minute averages of the retrieved Xgas values of the reference EM27/SUN ("tb") from those obtained from the other instruments. The only correction applied to the results at this point is the airmass-dependent correction. We calculate the median bias for Xgas between EM27/SUN pairs at each measurement location and look for any changes in the biases over time. Median biases in Xgas between each EM27/SUN and the reference instrument at each measurement station are shown in Figure 6. The error bars are calculated by taking the median absolute deviation (MAD) in the biases.

The range of the biases are upto 0.2 ppm in $XCO_2$, 4 ppb for $XCH_4$ and 0.8 ppb for XCO. In a study by Frey et al. (2019b), biases between 30 EM27/SUNs were evaluated and a scale factors of 0.999-1.0004 for XCO2 and 0.9975-1.0026 for $XCH_4$ were found. This would be equivalent to an average bias of 0.6 ppm for $XCO_2$ for an average DMF of 400 ppm and 9 ppb for $XCH_4$ for an average DMF of 1840 ppb. We observe smaller biases between our instruments, which might be expected as we only compare 4 EM27/SUNs.

The maximum change in EM27/SUN instrument-to-instrument biases in Xgas over time are calculated by taking the difference between the minimum and maximum bias in each EM27/SUN Xgas pair at all measurement locations. For both GGG2014 and GGG2020, the variability in the bias (MAD) calculated between retrievals from any EM27/SUN and the reference instrument at each measurement location is less than 0.1 ppm for $XCO_2$, 0.6 ppb for $XCH_4$ and 1 ppb for XCO.

Maximum site to site variation in the median biases with respect to the reference instrument are 0.07(0.08 for GGG2020) ppm for $XCO_2$, 0.47(0.45 for GGG2020) ppb for $XCH_4$ and 1.0(1.3 for GGG2020) ppb for XCO that are on the same order of magnitude as the maximum MAD in the biases across all sites. This reassures us that the instrumental drift or change in alignment is small throughout the campaign (Table 4).

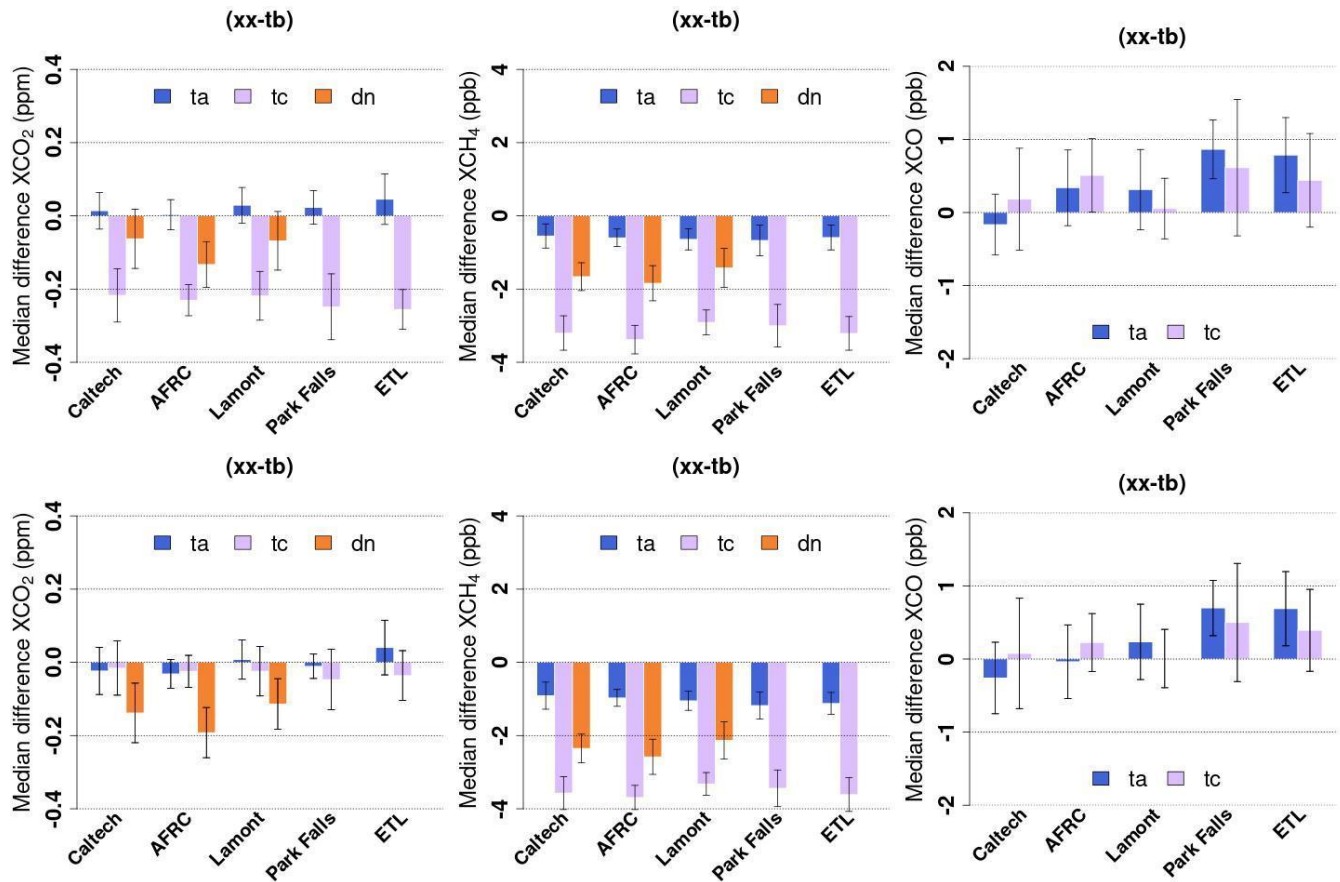

**Figure 6.** These figures show the median differences between the 10-minute averaged EM27/SUN $XCO_2$ (in ppm), $XCH_4$ and XCO (both in ppb) from coincident EM27/SUN measurements collected throughout the 2018 field campaign at each TCCON site before applying the instrument-to-instrument bias correction. The colour bars show the median biases of each EM27/SUN ("ta", "tc" and "dn") relative to the "tb" EM27/SUN instrument at each stop during the field trip. Top panel shows GGG2014 and the bottom panel GGG2020 results respectively. The error bars on each colour bar represent the median absolute deviation in the 10-minute averaged biases.

**Table 4.** This table lists the largest variations observed in biases measured by each EM27/SUN pair throughout the 2018 campaign. The maximum variation in the biases are calculated by taking the difference between the minimum and maximum 10-minute averaged biases in each EM27/SUN Xgas pair at each station. The maximum MAD is the maximum of all the MADs in the 10-minute averaged differences from each pair of EM27/SUNs at each station. This table shows these values for retrievals performed using GGG2014 and using GGG2020.

|  | GGG2014 | | GGG2020 | |
|---|---|---|---|---|
|  | max change in bias | max MAD | max change in bias | max MAD |
| $CO_2$ (ppm) | 0.07 (0.02%) | 0.09 | 0.08 (0.02%) | 0.08 |
| $CH_4$ (ppb) | 0.47 (0.03%) | 0.59 | 0.45 (0.02%) | 0.58 |
| CO (ppb) | 1.0 (1.1%) | 0.8 | 1.3 (1.4%) | 0.8 |

### 3.1.3 EM27/SUN scaling to the WMO trace gas scale

TCCON measurements have been scaled with respect to column-integrated vertical profiles measured by in situ instruments either collected on board aircraft or on balloon-based platforms. Similarly for the EM27/SUNs, we derive the airmass-independent correction factors (AICF) to scale the EM27/SUN measurements by integrating measured AirCore profiles that were collected during our campaign. We follow the method described in Wunch et al. (2010) and compare the integrated profiles with the EM27/SUN retrieved Xgas values. We force the regression line through origin and use the York et al. (2004) linear regression method, which accounts for the measurement errors in both the EM27/SUN and AirCore measurements to calculate the slope.

The relationship between the integrated AirCore profiles and the EM27/SUN measurements are presented in Figure 7 and Table 5, for both GGG2014 and GGG2020. The slopes are expected to be different between the two GGG versions due to the spectroscopic changes to the line shape models (Mendonca et al., 2016, 2017, 2019). We can see one data point measured at AFRC is an outlier for $XCH_4$. This is the only AirCore measurement that was performed early in the morning (14:00 UTC, 7:00 local time). The rest of the launches were performed around noon (11 am - 2 pm). Thus the difference could originate from the different slant column observed by the EM27/SUN compared to the AirCore path.

Each data point in Figure 5 represents a 1 hour EM27/SUN average centred on the average time over the descent of the AirCore (between maximum altitude and the surface). This is plotted against the mean of the averaging kernel-smoothed column integrated AirCore values. Extending the EM27/SUN averaging period to 2 hours did not change the obtained regression slope within the uncertainty.

Errors on the EM27/SUN values are derived by taking twice the standard deviation over the 1 hour period. Errors on the AirCore values originate from multiple sources: 1) Errors associated with the Picarro gas analyzer. 2) Errors in estimating the height at which each measurement point was collected. 3) Errors in estimating the profile above the AirCore ceiling altitude using GGG apriori profiles. 4) Errors associated with atmospheric variability due to different AirCore pathways. The overall error of AirCore is calculated by summing, in quadrature, these sources of error. The average analyzer error is 0.06 ppm for $CO_2$ , 0.9 ppb for $CH_4$ and 3 ppb for CO. The errors associated with altitude error were calculated by shifting the AirCore profile upward and downward by using the altitude error associated with each gas at each level and ceiling altitude error was calculated by integrating the profile above the AirCore ceiling assuming an error of 1 km. The errors due to altitude error and ceiling altitude error are negligible with orders of magnitude smaller than $10^{-4}$ % for $CO_2$, $10^{-3}$ % for $CH_4$ and 0.01 % for CO. The largest source of error is the error due to atmospheric variability and we took a conservative approach and use the maximum difference in column integrated values of AirCores launched at the same time to account for it. Errors due to atmospheric variability are 0.3 ppm for $CO_2$, 2.1 ppb for $CH_4$ and 8 ppb for CO.

TCCON has decided to set the GGG2020 XCO AICF to 1 due to concerns over drifts in the gas tanks used for calibrating in situ XCO measurements. Similarly, we choose a AICF of 1 for XCO to be consistent, though we recognize it might need to be updated in the future.

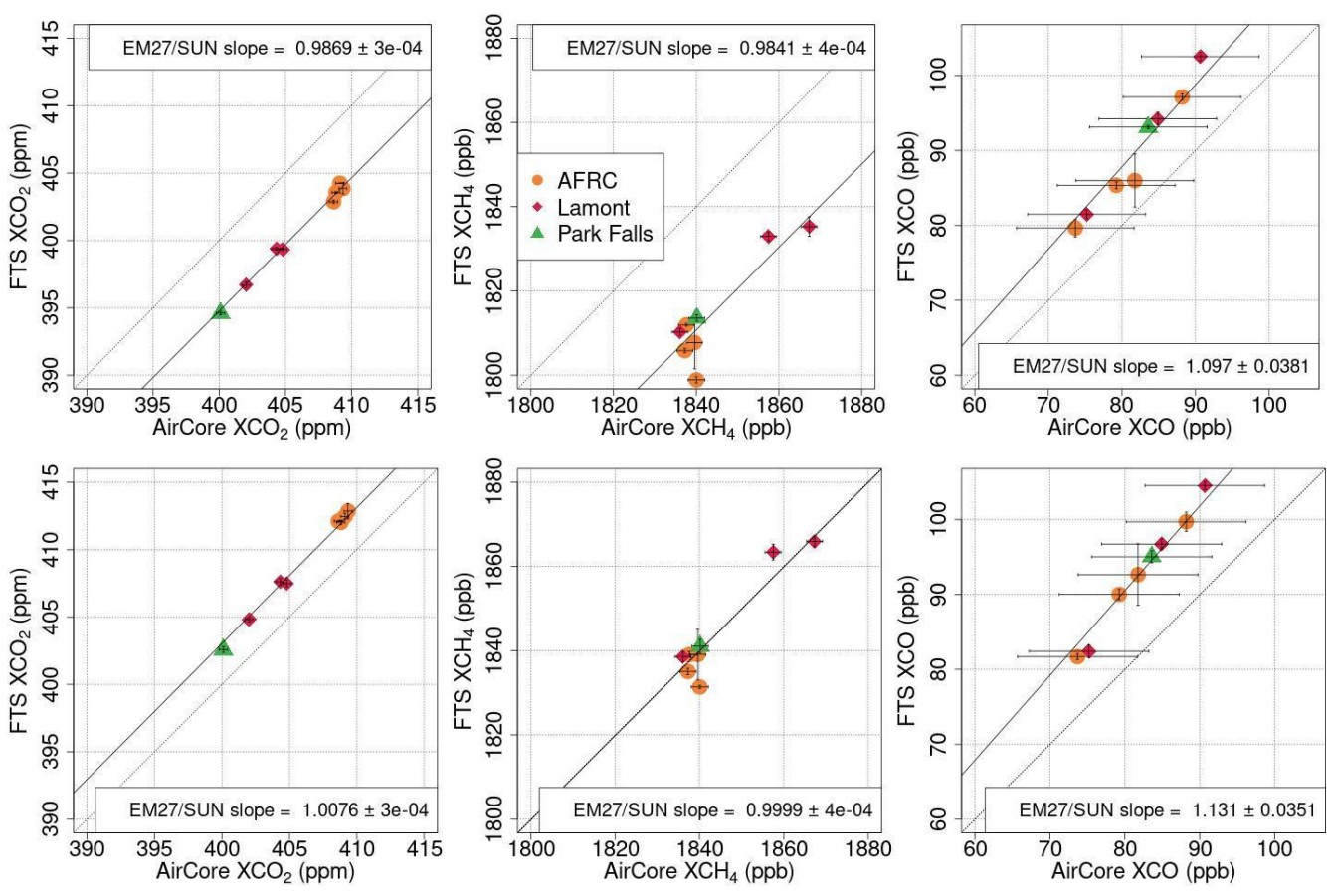

**Figure 7.** Correlation plots, regression line slopes and errors of Xgas from the EM27/SUNs ("ta" at AFRC remote site and "tb" at Lamont and Park Falls) and integrated column of AirCores collected at three of the TCCON sites. Top panel using GGG2014 for the retrieval and bottom panel using GGG2020.

**Table 5.** Airmass-Independent Correction Factors (AICF) used for TCCON and EM27/SUNs for GGG2014 and GGG2020.

|  | TCCON GGG2014 AICF | TCCON GGG2020 AICF | EM27/SUN GGG2014 AICF | EM27/SUN GGG2020 AICF |
|---|---|---|---|---|
| $XCO_2$ | 0.9898 | 1.0101 | 0.9869 | 1.0076 |
| $XCH_4$ | 0.9765 | 1.0031 | 0.9840 | 0.9999 |
| $XCO$ | 1.0672 | 1.0 | 1.0965 | 1.0 |

## 4 Evaluation of TCCON biases against the EM27/SUNs and Discussion

The data collected during the summer 2018 campaign are presented as a time series in Figure 8. We have included Xluft in Figure 8 which we use as a quality check to monitor the stability of the measurements. Xluft is defined as $Xluft = 0.2095 \times V_{\text{dry air}}/V_{\text{O}_2}$ where $V$ indicates a column density (in molecules per cm$^2$). $V_{\text{dry air}}$ is calculated from measured surface pressure at the time of measurement. Xluft is expected to be 1 with little variation. The observed variations in Xluft during the road trip remain within 0.5% of 1 for the EM27/SUNs, indicating good stability of the instrument retrievals. Table 6 shows the average Xluft values measured by TCCON and by the reference EM27/SUN at each measurement station are presented. The largest variations in Xluft are at Park Falls, which could be caused by the clouds that frequently interrupted measurements, leading to sparser data. At East Trout Lake, larger scatter is visible in XCO$_2$, XCH$_4$, and XCO, likely a real atmospheric effect caused by the forest fire plumes passing overhead (Statistics Canada, 2019).

**Table 6.** Average Xluft values measured by TCCON (125 HR) and the reference EM27/SUN (tb)

| TCCON Site | Average TCCON Xluft | Average EM27/SUN Xluft (tb) |
|---|---|---|
| Caltech (ci) | 0.9993 | 1.0020 |
| AFRC (df) | 0.9979 | 1.0014 |
| Lamont (oc) | 0.9990 | 1.0022 |
| Park Falls (pa) | 1.0028 | 1.0013 |
| East Trout Lake (et) | 1.0002 | 1.0040 |
| Eureka (eu) | 1.0005 | 1.0066 |

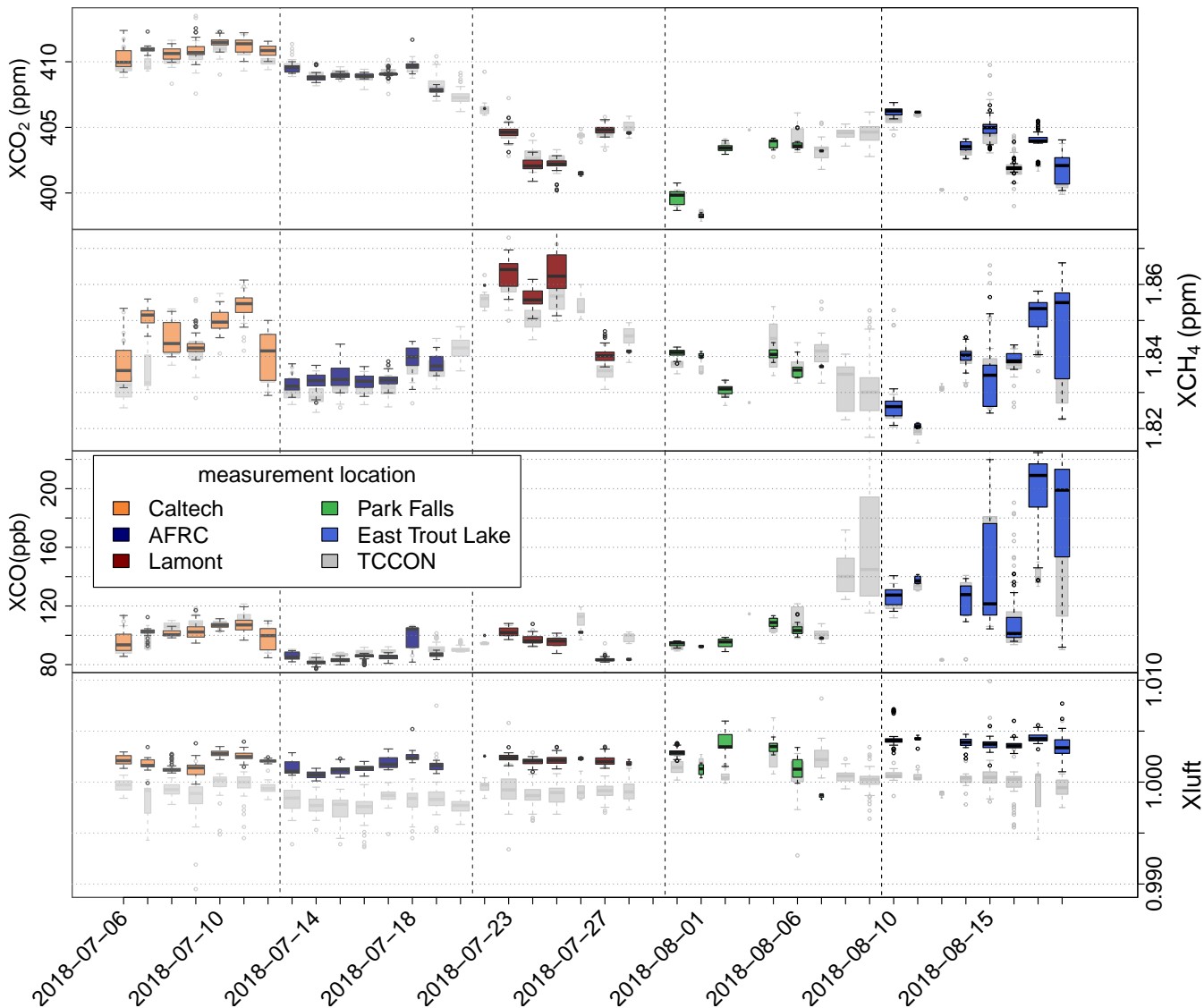

**Figure 8.** Timeseries of EM27/SUN ("tb") retrieved $XCO_2$, $XCH_4$, XCO and Xluft in colour and co-located TCCON in transparent grey during the summer 2018 campaign (GGG2020). Different sites are highlighted by a different colour. Vertical dashed lines represent the dates in which the instruments were moved to the next TCCON station. The boxplot width is proportional to the number of observations collected in each day. The line inside each box represents the median value for each day and the bars show the daily range excluding the outliers. Outliers are depicted as dots.

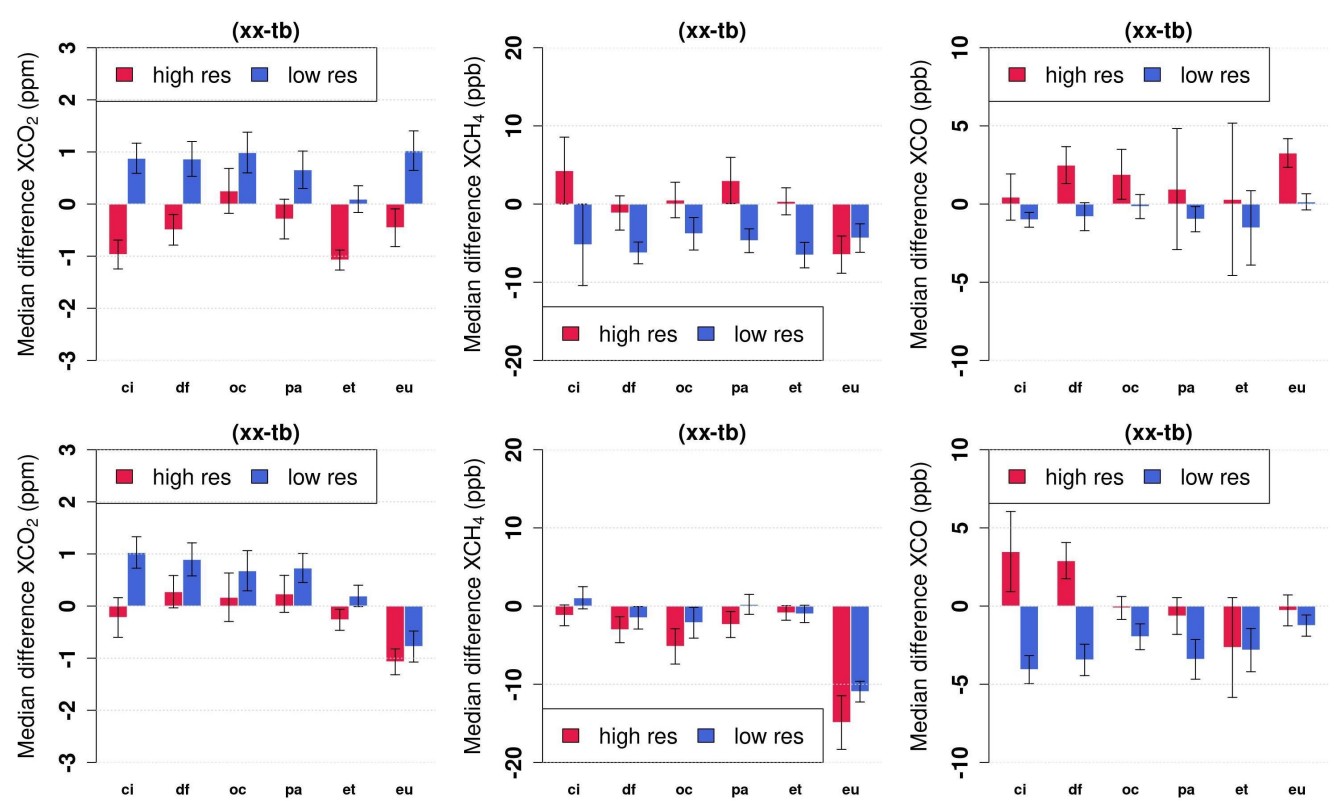

**Figure 9.** These figures show the differences between the 10-minute averaged EM27/SUN and TCCON $XCO_2$ (in ppm), $XCH_4$ and XCO (both in ppb) from coincident measurements collected throughout the 2018 field campaign and 2020 Arctic campaign. The colour bars show the median biases relative to the reference EM27/SUN ("tb") at each TCCON station. The error bars on each colour bar represent the median absolute deviation in the 10-minute averaged biases. Top panel shows GGG2014 and the bottom panel GGG2020 results respectively.

**Table 7.** This table lists the maximum EM27/SUN-TCCON and EM27/SUN-Low-resolution (LR) TCCON biases throughout the 2018 campaign. The maximum biases are calculated by taking the difference between the minimum and maximum 10-minute averaged biases between the EM27/SUN and each TCCON site. The maximum MAD is the maximum of all the MADs in the 10-minute averaged EM27/SUN-TCCON biases. This table shows these values for retrievals performed using GGG2014 and using GGG2020. Maximum EM27/SUN-TCCON biases where Eureka data from 2020 Arctic campaigns are included, are shown in brackets.

| | GGG2014 | | | | GGG2020 | | | |
| --- | --- | --- | --- | --- | --- | --- | --- | --- |
| | TCCON | | LR TCCON | | TCCON | | LR TCCON | |
| | max bias | MAD | max bias | MAD | max bias | MAD | max bias | MAD |
| $XCO_2$ (ppm) | 1.33 (1.33) | 0.53 | 0.89 (0.93) | 0.39 | 0.53 (1.35) | 0.47 | 0.83 (1.80) | 0.38 |
| $XCH_4$ (ppb) | 5.4 (10.8) | 4.3 | 2.7 (2.7) | 2.0 | 4.3 (14.0) | 2.2 | 3.2 (12.0) | 2.0 |
| XCO (ppb) | 2.2 (3.0) | 4.9 | 1.4 (1.7) | 2.2 | 6.1 (6.1) | 3.1 | 2.1 (2.8) | 1.4 |

To evaluate the biases in Xgas between TCCON and the EM27/SUNs, we compare coincident 10-minute averaged Xgas values from TCCON instruments and EM27/SUNs. On average, there are 65 EM27/SUN spectra and four TCCON spectra in each 10-minute average. Assuming measurements are dominated by Gaussian noise, variability in the EM27/SUN averages will be reduced by a factor of 8, and variability in the TCCON averages will be reduced by a factor of 2 relative to single spectrum measurements. The average biases obtained at all the sites are compared to each other to further evaluate site-to-site biases in TCCON.

Differences in the spectral resolutions can lead to differences in the retrieved Xgas values because high- and low-resolution measurements have different vertical sensitivities (column averaging kernels). In order to eliminate potential biases due to differences in resolution, we have also performed a comparison with truncated high-resolution 125HR interferograms to match the EM27/SUN resolution of 0.5 $\mathrm{cm}^{-1}$ (maximum OPD of 1.8 cm). The truncated spectra are then processed with GGG again to retrieve Xgas values. The 10-minute-averaged low-resolution 125HR Xgas values are also compared to EM27/SUN Xgas values.

Figure 9 presents the median bias between the 10-minute-average Xgas values from the reference EM27/SUN ("tb") and each TCCON instrument. The error bars are calculated using the median absolute deviation (MAD) in the 10 minute averaged biases. The red and blue bars show the differences when high-resolution (red) and low-resolution (blue) 125HR spectra are compared with the EM27/SUN measurements. The same ADCF and AICF used for the EM27/SUNs are applied to the truncated 125HR low-resolution Xgas retrievals. A summary of maximum site-to-site biases and MAD in the biases for each gas is presented in Table 7. Eureka comparisons are an outlier, particularly for GGG2020 $XCO_2$ and $XCH_4$, and so we exclude the Eureka data in the following results, discussing Eureka separately in Sect. §4.1.

For GGG2014, comparing the nominal (high-resolution) TCCON retrievals to the EM27/SUN retrievals results in maximum differences in site-to-site biases of 1.3 ppm for $XCO_2$, 5.4 ppb for $XCH_4$ and 2.2 ppb for XCO. When using low-resolution 125HR spectra, the biases reduce to 0.9 ppm for $XCO_2$, 3 ppb for $XCH_4$ and 1.4 ppb for XCO. The MAD in the differences improves for the low-resolution 125HR spectra suggesting better consistency between the two instrument retrievals.

For GGG2020, comparing nominal TCCON retrievals to EM27/SUN retrievals results in maximum site-to-site biases of 0.53 ppm for $XCO_2$, 4.3 ppb for $XCH_4$ and 6 ppb for XCO. The TCCON-EM27/SUN bias shows improvement in $XCO_2$ and $XCH_4$ compared to GGG2014 but not for XCO. When we compare the EM27/SUN retrievals to the low-resolution TCCON retrievals, the site-to-site biases increase to 0.83 ppm for $XCO_2$ and decrease to 3.2 and 2.1 ppb for $XCH_4$ and XCO, respectively. The largest differences between nominal TCCON XCO and EM27/SUN XCO are at Caltech and AFRC, which are both close to the urban Los Angeles region where the a priori profile, which is taken from the GEOS-FPIT model, is known to overestimate near surface emissions of CO. Excluding the Caltech and AFRC data from the XCO bias calculation leads to a maximum bias of 2.5 ppb. Because the differences in XCO between the TCCON instruments and the EM27/SUNs are smaller at low-resolution, this suggests that about half of the XCO bias likely originates from errors in the GGG2020 a priori profile shape of CO and the different sensitivities of the low-resolution and high-resolution instruments.

Generally, differences we see between TCCON measurements at sites other than Eureka are small, on the same order of magnitude of the TCCON error budget and could point to instrument alignment differences or other retrieval related uncertain-

ties.

## 4.1 Eureka

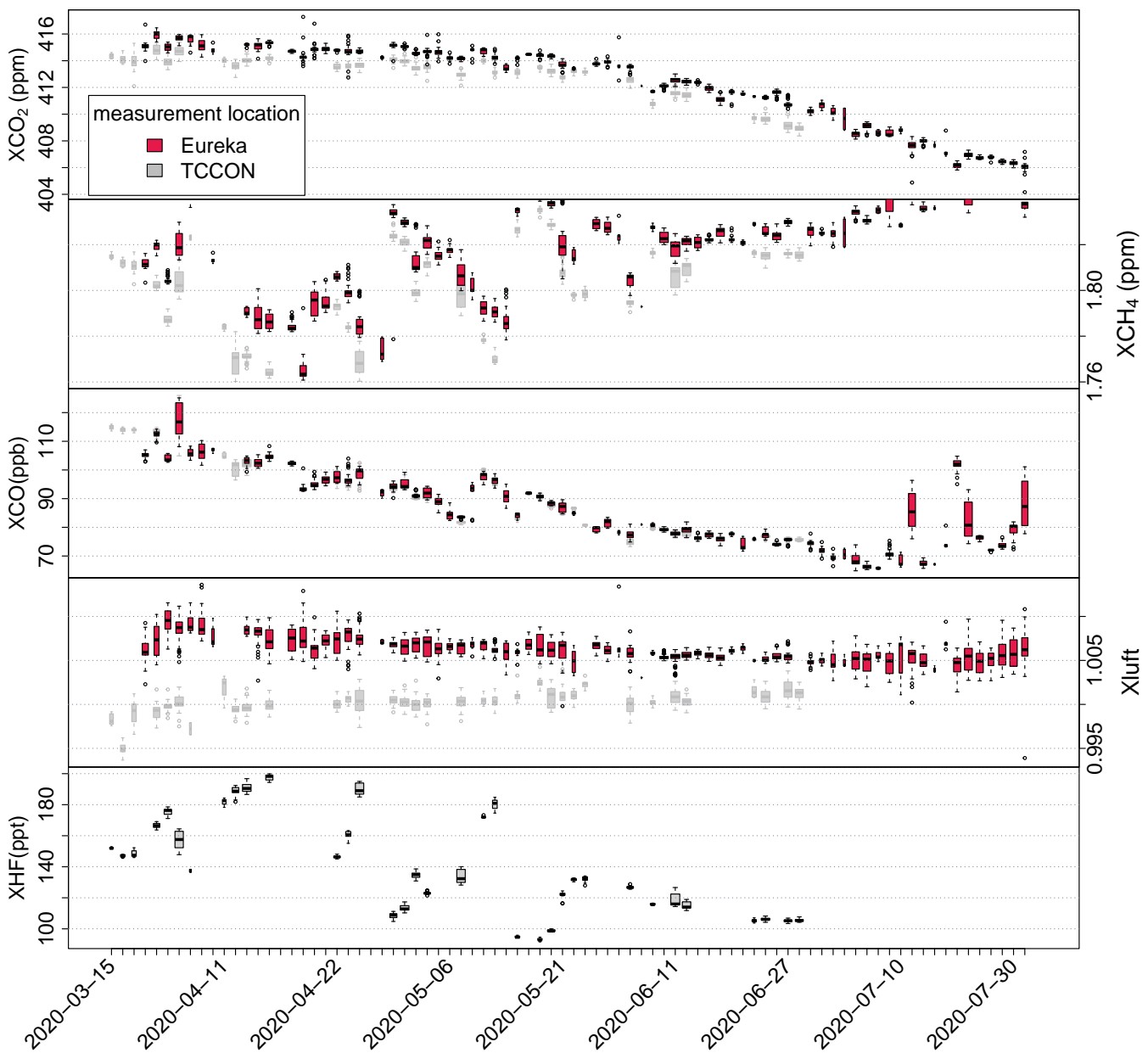

**Figure 10.** Timeseries of EM27/SUN (tb) and co-located TCCON retrieved XCO₂, XCH₄, XCO, Xluft and XHF (TCCON only) during the spring and summer 2020 Eureka campaign (GGG2020). The boxplot width is proportional to the number of observations collected in each day. The line inside each box represents the median value for each day and the bars show the daily range excluding the outliers. Outliers are depicted as dots.

Figure 10 shows the Xgas timeseries at Eureka in spring-summer 2020. The dataset is long enough that it allows us to observe part of the seasonal cycle in Xgas values. Abrupt changes in XCH$_4$ are visible around March 20th, May 1st and May 16th. TCCON retrievals of XHF (hydrogen fluoride), a gas only present in the stratosphere, also show an abrupt change at the same time. This indicates that the tropopause height was changing, likely caused by the position of the polar vortex (Bognar et al., 2021). There are enhanced XCO events visible that could also indicate long-range transport of forest fire plumes in the summer (e.g., Lutsch et al., 2016).

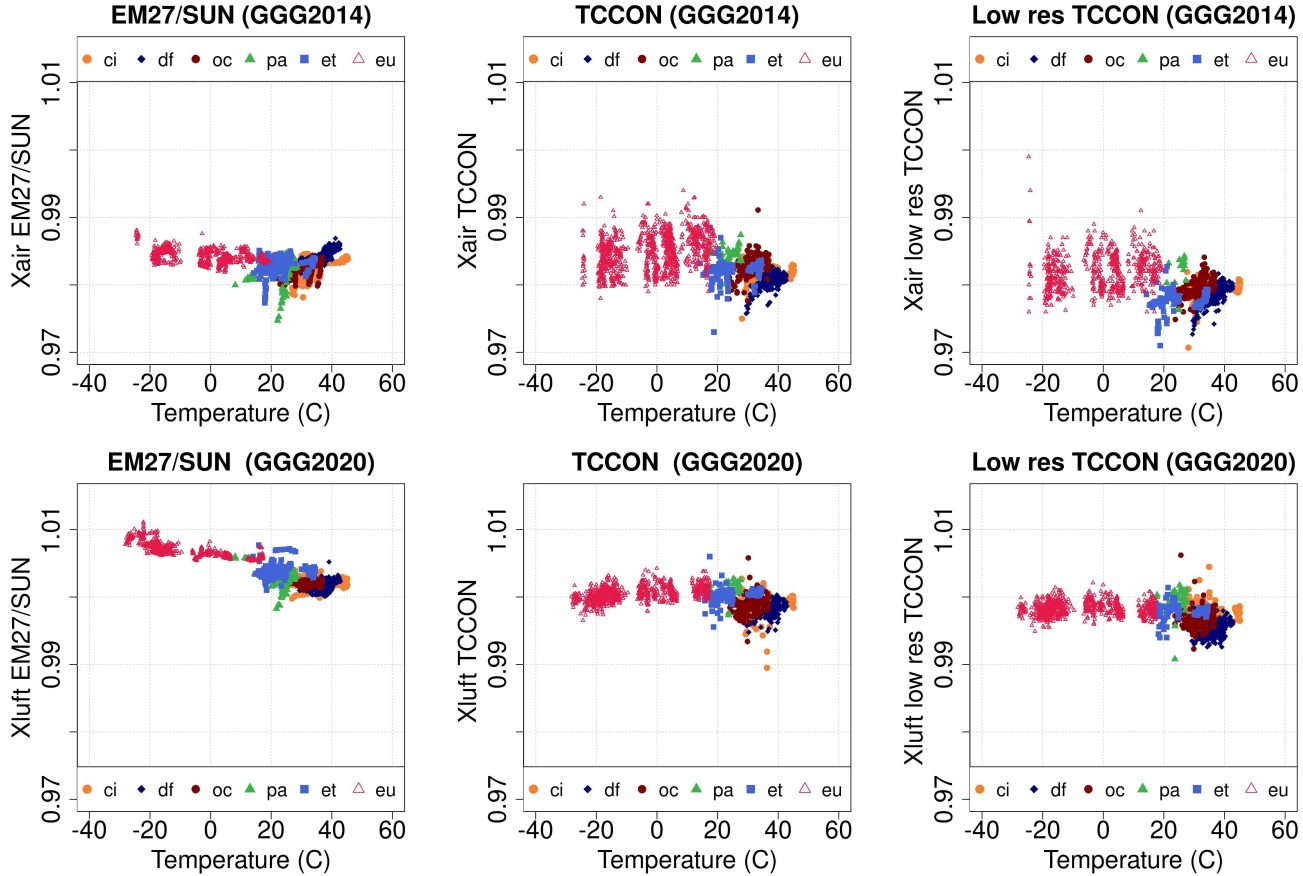

**Figure 11.** Xair (GGG2014) and Xluft (GGG2020) against surface temperature for the EM27/SUNs (left panel), TCCON (middle panel) and low-resolution TCCON (right panel).

The data collected at Eureka show a clear difference between the EM27/SUN measurements and the TCCON measurements for GGG2020 XCO$_2$ and XCH$_4$ retrievals, both for high-resolution and low-resolution comparisons (Figure 9 and Table 7). Biases between TCCON Eureka retrievals and the EM27/SUNs has increased from GGG2014 to GGG2020 from 0.4 ppm to 1.1 for XCO$_2$ ppm and from 6.2 ppb to 12.2 ppb for XCH$_4$ and has reduced from 3.2 ppb to 1.6 ppb for XCO. Using the

low-resolution TCCON spectra, the bias reduces from 1 ppm to 0.7 ppm for $XCO_2$ and increases from 4 ppb to 8.8 ppb for $XCH_4$ and from 0.2 ppb to 2.3 ppb for XCO.

One obvious difference when measuring direct sunlight at Eureka compared to the other locations in this study is the high SZA values encountered that could exacerbate any airmass-dependent biases. However, we did not observe airmass dependence in the biases, and limiting the data to lower SZA values does not improve the biases. Moreover, using GGG2014, the Eureka biases were generally in good agreement with the rest of the sites.

Another significant difference between Eureka and other sites is the atmospheric temperature: Eureka's surface temperature during the March–July period ranged from $-25°C$ to $+20°C$. The $O_2$ line widths are affected by both temperature and water concentrations in the atmosphere, and an empirical update to the $O_2$ spectroscopy was developed using high-resolution data from multiple TCCON stations. This updated spectroscopy is used in the GGG2020 algorithm, but not in the GGG2014 algorithm. In Figure 11, we plot the GGG2014 Xair and GGG2020 Xluft against surface temperature for the EM27/SUNs, and the TCCON high and low-resolution measurements. This figure shows that the EM27/SUN Xluft temperature dependence becomes significantly worse in GGG2020 than it was in GGG2014. Interestingly, the same temperature-dependent effect at Eureka is not seen in the low-resolution TCCON measurements, indicating that the discrepancy in Xgas biases between TCCON and EM27/SUNs at Eureka originates from the spurious temperature dependence in the EM27/SUN data collected at Eureka, and not the TCCON measurements. Figure 11 also shows that the temperature dependence at the other sites are very similar between the EM27/SUN, the high-resolution, and low-resolution TCCON measurements.

## 5    Conclusions and future work

During the 2018 summer campaign, we visited five TCCON sites in North America with four EM27/SUN portable FTS instruments. The change in 10-minute-averaged biases between the EM27/SUNs Xgas retrievals remained consistent within the respective uncertainty of measurements of each gas as we moved the EM27/SUNs from one site to the other via ground transportation. We find negligible EM27/SUN instrumental drift over time. Therefore, we see great promise for the EM27/SUN instruments to be used to investigate site-to-site biases in TCCON that are larger than 0.5 ppm for $XCO_2$, 4 ppb for $XCH_4$, and 2 ppb for XCO, comparable with TCCON GGG2020 error budget.

To use EM27/SUNs as a "travel standard" for indirect comparison of TCCON stations we suggest using the latest version of EGI2020 (Hedelius and Wennberg, 2023b) with the GGG2020 retrieval algorithm. The following three steps are automatically applied in the post-retrieval data processing by EGI2020 to ensure that all the EM27/SUNs are on the same scale:

1. Apply EM27/SUN-specific airmass dependent corrections (ADCFs) presented in this work to the window-averaged Xgas retrievals.

2. Apply a constant bias on Xgas based on side-by-side comparisons against the reference EM27/SUN (currently JPL's EM27/SUN ("dn"), sn: 42). This needs to be specified in an input file the first time.

3. Apply EM27/SUN-specific scaling to the WMO trace gas standard scale (AICF) that we have derived based on comparisons with coincident AirCore profiles.

For historical algorithm comparisons with GGG2014 we suggest using EGI2014.5 (Hedelius and Wennberg, 2023a) or later, which will apply our recommended ADCF and AICF scalings.

In addition to the summer 2018 road trip, in spring and summer 2020, we sent one instrument to Eureka, the Canadian Arctic TCCON site, and we collected side-by-side measurements against the TCCON station located there. These long-term measurements provide a valuable dataset to assess the EM27/SUN-TCCON biases across seasons and under different atmospheric

conditions. More measurement days at other sites and in different seasons would provide an opportunity to better investigate the origin of the observed site-to-site biases. From the measurements at Eureka, we discovered a temperature-dependent bias in the EM27/SUN GGG2020 retrievals that were not present in the GGG2014, and this is an issue that requires further study. For that reason, we do not include the Eureka measurements in our reported TCCON site-to-site biases.

We calculate TCCON site-to-site biases indirectly by using the EM27/SUN as a "transfer standard" while performing side-

by-side measurements at each site. For $XCO_2$, when using GGG2014, the site-to-site biases are significantly larger than the variability in the bias (0.5 ppm), but when degrading the TCCON spectra to match the EM27/SUN resolution, the site-to-site biases become more consistent everywhere except at ETL, which was significantly affected by wildfire smoke during the campaign. When using GGG2020, the site-to-site biases in the high resolution TCCON agree everywhere within the expected variability. However, employing the low resolution TCCON caused the maximum bias to rise, contrary to what was anticipated.

For $XCH_4$, when using GGG2014, the significant site-to-site biases improve when comparing the EM27/SUNs to low-resolution TCCON spectra. The maximum $XCH_4$ site-to-site bias decreases when using GGG2020, and switching to low-resolution spectra decreases the bias further.

For XCO, the variability is generally of the same order of magnitude as the biases (2–5 ppb) with the exception of the GGG2020 retrievals at Caltech and AFRC, where biases are significantly larger than at the rest of the sites. This is likely due

to profile shape errors in the new GGG2020 a priori profiles at these two sites. Further investigation is required to assess and improve on the shape of the a priori profiles in urban and other high emission areas.

Frey et al. (2019b) performed comparisons between an EM27/SUN at Karlsruhe Institute of Technology (KIT) and the KIT TCCON instrument and found a scaling factor of 1.0098 for $XCO_2$ for high resolution TCCON and 1.0014 for low resolution TCCON retrieval equivalent to 4 ppm and 0.6 ppm respectively. For $XCH_4$, a scaling factor of 1.0072 was found for high

resolution TCCON and 0.9997 for low resolution TCCON equivalent to 13 ppb and 0.5 ppb respectively. In Frey et al., the high resolution TCCON spectra were processed using GGG2014, whereas the low resolution TCCON and the EM27/SUN retrievals were performed using a different retrieval algorithm (PROFFIT version 9.6), which could explain the larger bias found between the EM27/SUN and TCCON in the Frey et al. work, compared to this study.

Site-to-site biases can originate from instrument misalignments such as ILS or pointing errors, or the retrieval algorithm

and a priori profile estimations. Comparing the EM27/SUN measurements with the low-resolution TCCON spectra eliminates biases caused by errors in the a priori profile shapes, and biases caused by differences in the spectral resolution, because the vertical sensitivities of the low-resolution retrievals will be the same. However, comparisons of EM27/SUNs with GGG2020

low-resolution TCCON retrievals did not improve the site-to-site biases for $XCO_2$. We see value in continuing to retrieve Xgases from low-resolution TCCON measurements and compare such retrievals to high resolution TCCON retrievals over extended periods of time to further evaluate the quality of low-resolution TCCON retrievals.

## Appendix A:  Filtering - Pointing errors

In the processing of the measured solar spectra, absorption lines from solar atmospheric gases are modeled and fitted. However, if the instrument points away from the sun's spin axis, these absorption lines will move as a consequence of the Doppler effect. The retrieval algorithm estimates the stretch caused by the pointing error, however when the offset from the centre is large, the spectral fits will be poor and the retrieved gas columns could be biased. The largest influence of this mis-pointing effect is observed in XCO retrievals because the solar lines overlap the telluric CO lines (Wunch et al., 2011). Thus we remove the spectra with large solar shift values. In addition to quality flags assigned by GGG post-processing, we have added another quality flag that filters out large deviations in solar gas shift values. The spectra with the solar gas shifts beyond 2 sigma of the daily median are removed. Figure A1 demonstrates how the filtering is performed in a typical measurement day.

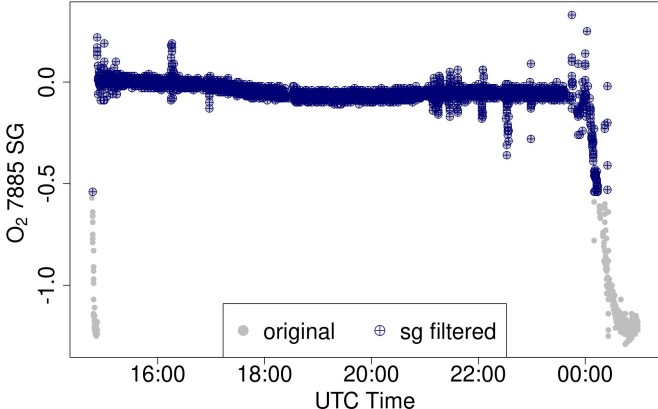

**Figure A1.** EM27/SUN solar gas shift (SG) values before (grey) and after the additional filtering (blue). Units of SG are ppm of the spectral points spacing.

## Appendix B:  Pressure calibration

The TCCON pressure sensors (Setra pressure transducer and GE-8100 pressure sensor) that are already calibrated against the standard Digiquartz sensor are used for EM27/SUN retrievals. Comparisons of TCCON pressures against the standard Digiquartz pressure sensor during the campaign at each TCCON station are shown in Figure B1. At AFRC and Lamont where the EM27/SUN was deployed at a different height than the TCCON mirror, the pressure was adjusted slightly ($0.1 - 0.2$ hPa) to match the standard pressure sensor values. At Park Falls, a Vaisala WXT356 sensor was used for both TCCON and EM27/SUN

retrievals instead of the TCCON pressure sensor due to the sensor instability. The Vaisala pressure sensor had already been calibrated against the Digiquartz standard.

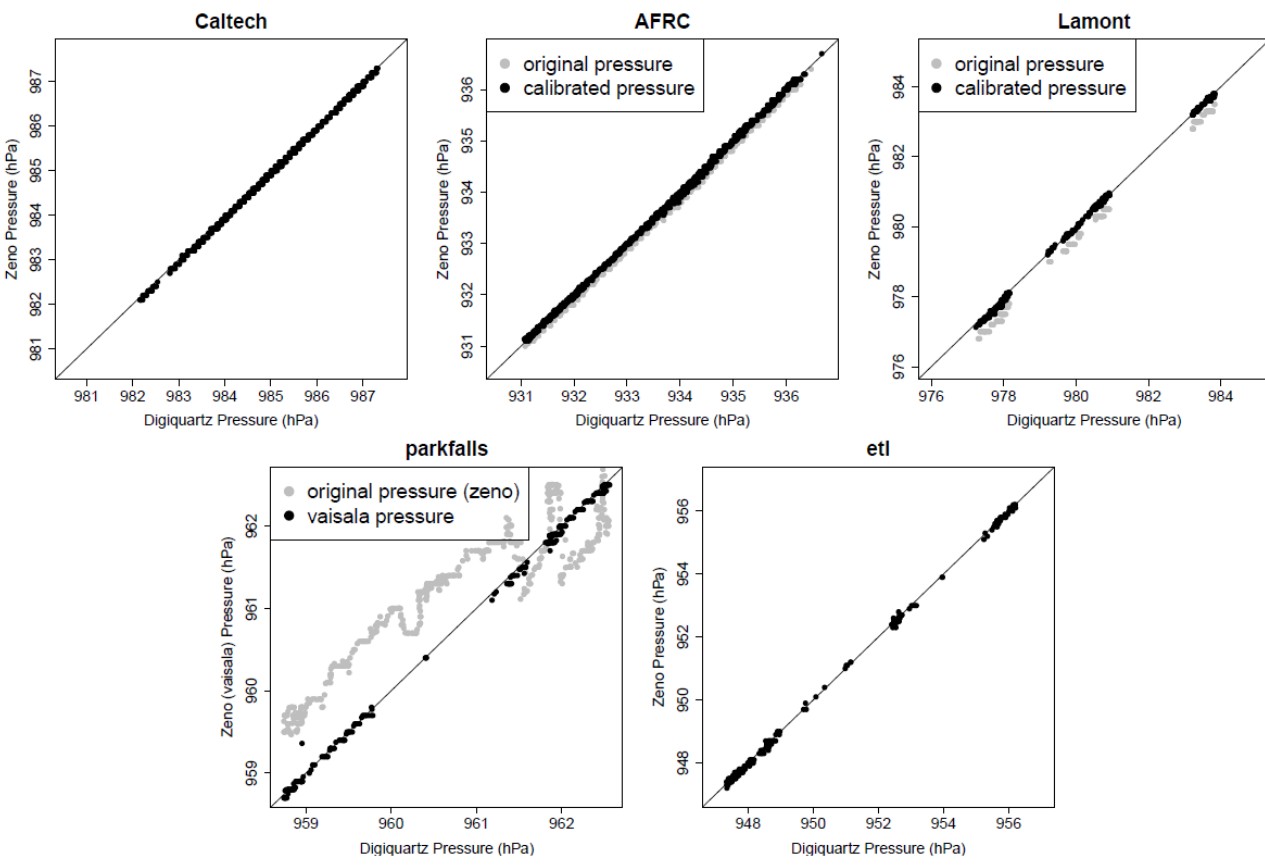

**Figure B1.** Surface pressure used for the EM27/SUN and TCCON retrievals compared against the standard Digiquartz pressure sensor. At AFRC and Lamont, a small bias was added to the EM27/SUN pressure to match the Digiquartz pressures. At Park Falls, Vaisala WXT536 pressures are also plotted.

*Author contributions.* **Nasrin Mostafavi Pak**: Wrote the manuscript. Performed EM27/SUN measurements. Performed EM27/SUN retrievals. Analysed FTS and AirCore data. Created visualizations. **Jacob Hedelius**: Development of the software suite EGI and EGI2020. Performed EM27/SUN measurements. Interpretation of the FTS data and comparisons. **Sébastien Roche**: Assisted with TCCON measurements at Eureka. Interpretation of the FTS data and investigation of differences between EM27/SUN and 125 HR at Eureka. **Liz Cunningham**: Performed EM27/SUN measurements. Assisted with interpretation of AirCore data. **Bianca Baier**: Performed AirCore measurements, provided the data, assisted with AirCore data interpretation. **Colm Sweeney**: Assisted with AirCore measurements. **Coleen Roehl**: Provided TCCON retrieval results and ILS results. **Joshua Laughner**: GGG2014 and GGG2020 development. Assisted with GGG2014 and GGG2020 analysis. **Geoffrey Toon**: GGG2014 and GGG2020 development. Assisted with GGG2014 and GGG2020 analysis. **Paul Wennberg**: Hosted

EM27/SUNs at Caltech. Provided feedback on the analysis. Development of GGG. Provided TCCON data. **Harrison Parker**: Performed EM27/SUN measurements. **Colin Arrowsmith**: Performed EM27/SUN measurements. **Joseph Mendonca**: Performed EM27/SUN measurements. **Pierre Fogal**: Assisted with EM27/SUN and TCCON measurements at Eureka. **Tyler Wizenberg**: Performed TCCON and EM27/SUN FTS measurements at Eureka. **Beatriz Herrera**: Performed TCCON and EM27/SUN FTS measurements at Eureka. **Kimberly Strong**: Hosted EM27/SUNs at Eureka. Provided Eureka TCCON data. **Kaley A. Walker**: Designed the arctic campaign and funding for the Eureka portion of the study. **Felix Vogel**: Provided EM27/SUN instrumentation and feedback on the analysis. **Debra Wunch**: Study design and funding. TCCON retrievals and analysis for ETL and all low-resolution TCCON measurements. GGG2014 and GGG2020 development. All coauthors provided feedback on the manuscript.

*Code and data availability.* The TCCON GGG2014 and GGG2020 retrievals at nominal resolution are available through the Caltech library and can be downloaded from https://tccondata.org/. EM27/SUN retrievals are available through the Borealis data archive: https://borealisdata.ca/dataverse/em27sun. The latest version of AirCore profiles are available at NOAA Global Monitoring Laboratory Data Repository: https://gml.noaa.gov/ccgg/arc/?id=144. The version used in this manuscript (v20181022) will be made available upon request. The EM27/SUN GGG interferogram processing suite (EGI) is available through the Caltech library: https://data.caltech.edu/records/25tve-4h822.

*Competing interests.* The authors declare that they have no competing interests.

*Acknowledgements.* Funding for the 2017 and 2018 field campaigns was provided by the New Researcher Award from the University of Toronto's Connaught Fund (NR-2015-16). Infrastructure funding was provided by the Canada Foundation for Innovation (35278), the Ontario Research Fund (35278), and Environment and Climate Change Canada, and analysis funding was provided by NSERC (RGPIN-2021-03525 and RGPAS-2021-00024). A portion of this research was carried out at the Jet Propulsion Laboratory, California Institute of Technology, under a contract with the National Aeronautics and Space Administration (80NM0018D0004).

The Canadian Arctic ACE validation campaigns are funded by the Canadian Space Agency (CSA), Environment and Climate Change Canada (ECCC), the Natural Sciences and Engineering Research Council of Canada (NSERC), and the Northern Scientific Training Program. CANDAC and PEARL are supported by the Atlantic Innovation Fund/Nova Scotia Research Innovation Trust, Canadian Foundation for Climate and Atmospheric Sciences, Canada Foundation for Innovation, CSA, ECCC, Government of Canada International Polar Year funding, NSERC, Ontario Innovation Trust, Polar Continental Shelf Program, and the Ontario Research Fund.

Special thanks to Jack Higgs, Sonja Wolter, and Timothy Newberger who helped with collection of AirCore profiles. AirCore funding source is NASA award number 80NSSC18K0898 .

We thank Benedikt Herkommer and David Griffith who provided helpful referee comments that significantly improved the clarity of this paper.

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
