# Peer review of "Using portable low-resolution spectrometers to evaluate TCCON biases in North America"

_EGUsphere, 2022_

## Author Comment (AC1)

**Authors' Response to Reviews of**

**Using portable low-resolution spectrometers to evaluate TC­CON biases in North America**

Nasrin Mostafavi Pak, Jacob Hedelius, Sébastien Roche, Liz Cunningham, Bianca Baier, Colm Sweeney, Coleen Roehl, Joshua Laughner, Geoffrey Toon, Paul Wennberg, Harrison Parker, Colin Arrowsmith, Joseph Mendonca, Pierre Fogal, Tyler Wizenberg, Beatriz Herrera , Kimberly Strong, Kaley A. Walker, Paul Wennberg, Felix Vogel, Debra Wunch
*Atmospheric Measurement Techniques,*
* * *
**RC:** *Reviewers' Comment*,     AR: Authors' Response,     ☐ Manuscript Text

We would like to thank the reviewer for their thoughtful comments that have helped to improved our manuscript substantially. The detailed response and corrections are given below.

**Reviewer #1**

**General Comments**

1. ## General Comment #1

RC: *In Section 2.3, you are dealing with ILS measurements. In Figure 2 you show the ILS measurements of the TCCON stations but you are not classifying them (are they good/bad?). They are not used for a further data analysis, I do not see a strong need to show them here. Is there an acceptance level? The variation of 1% found for the fluctuations in EM27/SUN modulation efficiency (ascribed to variable humidity and room temperature) seems way too high, Alberti et al., 2022 demonstrated a significantly better reproducibility (see Fig. 15 in this work). It is important to take the variable partial pressure of H2O into account when doing the analysis (which can be calculated from total pressure, path length and H2O column) and to do the analysis repeatedly until a self-consistent solution (for ILS, column, and partial pressure) is found. If this has been taken into account, I would suspect that there was a problem in coupling the light source to the spectrometer in a reproducible manner.*

AR: ILS results from TCCON and EM27/SUNs are used as a qualitative measure to ensure the stability of the instruments over the course of the road trip and ILS values are not currently used in the retrieval procedure. Regarding ILS measurements for TCCON, guidelines require that the modulation efficiency deviates less than 5% from 1.0, over 0 to 45 cm OPD. which is true for all the TCCON sites during our visit.

Regarding the EM27/SUN ILS measurements, we have followed the same procedure using partial pressure of H2O. We agree that the variability observed in Alberti et al., 2022 is significantly smaller, however there are multiple factors that could have caused a larger range for our measurements. In Alberti et al, all measurements are performed in one room using a single distance between the lamp and the FTS mirror. However during our road trip, we have performed ILS measurements in three different regions in North America with different laboratory conditions. We have also used 3 different lengths between the lamp and the FTS mirror and took the average of the 3 trials. This could contribute to the variability in the calculated ME. Further, these instruments underwent thousands of kilometers of travel, compared to the KIT reference instrument which

by design is not subjugated to as significant of movements. This movement could affect the ILS of the instruments.

We have added additional explanation in this section to clarify these points:

> Instrumental Line Shape (ILS) is a measure of the optical alignment of the instrument and imperfections in this alignment can cause biases in the retrievals. The ILS of an FTS can be described by two parameters: Phase Error (PE) and Modulation Efficiency (ME). For high resolution FTS instruments (TCCON) ILS is typically reported as function of optical path difference (OPD) and for the low resolution EM27/SUNs these are typically reported at the maximum optical path difference. ILS values are not implemented into the GGG2014 and GGG2020 retrieval algorithms and are only used to evaluate the instruments' alignment qualitatively. TCCON guidelines require that the modulation efficiency deviates less than 5% from 1.0, over 0 to 45 cm OPD. A modulation loss of 1% in the EM27/SUN causes a bias of 0.1% in $XCO_2$ and 0.15% in $XCH_4$ (Hedelius et al., 2016), therefore we aim to ensure that the EM27/SUN ME variations remain less than 1%.
>
> Because the EM27/SUNs were moved from one site to another, we evaluated their optical alignment by measuring the ILS of all the EM27/SUNs at various sites (Table 1). For EM27/SUNs, we use a method introduced by Frey et al. (2015). and further developed by Alberti et al. (2022), in which we collect spectra from an external lamp (Quartz Tungsten-Halogen Lamp, Thorlabs) in the laboratory and use LINEFIT (version 14.0) to derive the ILS parameters from $H_2O$ lines in the 7000 and 7400 cm $^{-1}$ spectral region (Frey et al., 2015). Our method differs slightly because we use three different distances between the lamp and the EM27/SUN, and we compute the standard deviation of the calculated ME across the three distances to evaluate the variability in the calculated ME.

**2. General Comment #2**

**RC:** *You derive the airmass-depended correction factors by comparing the measurements in the course of the day to the daily median, thereby assuming the variation is solely due to an airmass dependency. I am in doubt if this assumption is valid since it ignores intraday variability of GHGs, which, especially in a rural area like Toronto, seem reasonable for me to occur.*

AR: Although intraday variability in $CO_2$ is expected to some extent in urban areas such as Toronto due to peak traffic hours in the mornings and the afternoons, it's not regularly observed in our daily measurements. Based on measurements in multiple locations in Toronto, positive daily anomalies of larger than the standard deviation (0.6 ppm) occur mostly around solar zenith angles of 40–50 and around 14-16 UTC that is equivalent to 9 am- 11 am local time and are not symmetric around noon (See left panel of Figure 1 below). Therefore wouldn't interfere with the negative anomalies that occur at high solar zenith angles and are symmetric around noon (See right panel of Figure 1 below) and didn't have a significant effect on the final ADCF calculations. We collect fewer measurements in the early morning than in the late afternoon which skews the distribution. As can be seen in Figure 2, after applying the airmass correction, the negative anomalies have been eliminated effectively (right panel of Figure 2) and the enhancements due to urban emissions are not affected as strongly.

Figure 1: Before airmass correction

[Figure]

Figure 2: After airmass correction

[Figure]

We have added additional explanation in the text to clarify this point:

> To derive the airmass-dependent correction factors (ADCFs) for the EM27/SUNs, we use the long
> term record of measurements in Toronto from 2018 to 2021 with four EM27/SUNs ("ta", "tb", "tc" and
> "td" ) to calculate an average ADCF value for each gas. Although Toronto is under the influence of
> urban emissions, our analysis showed that the enhancements due to traffic emissions are not symmetric
> around noon and therefore would not interfere with the airmass dependent calculations.

**3. General Comment #3**

**RC:** *Next you are writing in line 281 you are using the same method as GGG2014 to apply the correction*
*factors. However, there is missing an explanation or at least a specific citation on how this is done in*
*GGG2014.*

AR: We have made necessary changes in the text to make this more clear:

> TCCON applies ADCFs for GGG2020 differently than in GGG2014. In the TCCON post-processing procedure, ADCF values are calculated for each retrieval window *before* averaging them (Laughner et al., 2020), whereas in GGG2014, ADCFs are calculated and applied for each gas after averaging different retrieval windows. Additionally, in GGG2020, $\theta_0$ and $p$ vary from window to window in order to best capture the airmass dependence, whereas in GGG2014 all gases used $\theta_0 = 13°$ and $p = 3$. For EM27/SUNs, we follow the same method as in GGG2014 and derive and apply ADCFs for each gas *after* averaging individual windows. It is therefore not possible to directly compare TCCON and EM27/SUN ADCF values for gases with multiple windows (i.e., $CO_2$ and $CH_4$); it is expected that they will be different.

We have also added the formula used to apply the airmass dependance correction:

> $$y_c = \frac{y_i}{[1 + \beta S(\theta_i)]} \qquad (1)$$
>
> where $\beta$ is the airmass dependent correction factor (ADCF) and $y_c$ is the airmass corrected Xgas value.

**4. General Comment #4**

RC: *You are using the median for the comparison of the 10-minute bins of the different spectrometer but you are using the mean value to calculate the 10-minute bins. What are the reasons for choosing the mean or the median for the different situations?*

AR: Over 10 minutes we expect the data from each instrument to be smooth, therefore the average and the median are not significantly different. To compare the two instruments, we use median over the entire data set to eliminate the possible outliers in the biases. The results wouldn't be greatly altered if we used the average.

**5. General Comment #5**

RC: *In chapter 3.1.2 you are describing how the maximum biases are calculated. However, this procedure is not clear to me. I understand you are taking the 10-minutes bins of the different spectrometers and then calculate the difference of the minimal and maximal bin each, even though they are not temporal coincident. This however, would include the variation of the XGas value to the maximum bias. Please clarify what is done there.*

AR: The aim of this section is to estimate biases between pair of EM27/SUNs at different measurement locations and compare them as we move from one site to the other to ensure it remains consistent. We agree that presenting the absolute bias is not as critical as the changes in the bias over time. Therefore we updated the text and the table 4 to present the changes in the biases. We took the minimum bias and maximum bias in each pair and took the difference to account for the worst case scenario, ie. what would be the maximum variation in the biases and show even in the worst case the variations in the bias are smaller than the medium absolute deviation calculated for the average biases calculated in each case.

**6. General Comment #6**

**RC:** *Lastly, you are writing in the introduction that you were taking a pressure sensor to the road trip to compare with the pressure measurement done on site. However, no detailed comparison is included in the paper. It would be nice to at least say a few words to the pressure comparison or better to show some results (maybe in a table?).*

**AR:** The digiquartz pressure sensor accompanied the EM27/SUNs and performed the measurements at the same height as the EM27/SUNs. As mentioned in Line 169, the slight differences in height of the EM27/SUNs and 125HRs in some of the TCCON sites led to minor differences in pressure. The offset was calculated based on the bias between the digiquartz pressure sensor and TCCON's pressure sensors that were previously calibrated against a similar pressure sensor. Therefore, we don't see the need to present the pressure comparison other than the cases we observed an offset due to height difference. We have also added an appendix to present the comparisons between the Digiquartz standard and TCCON pressure sensors at each site.

We made some changes to the corresponding paragraph to make it more clear how the standard pressure sensor was used.

> For the EM27/SUN retrievals, we use the pressure measurements recorded by the local TCCON weather stations which have already been calibrated against a Digiquartz pressure sensor, with the exception of Park Falls where we used the Vaisala WXT536 pressure data for both TCCON and EM27/SUNs, since the pressure measurements made by the TCCON pressure sensor were not stable during the campaign. In addition, at AFRC and Lamont we applied additional corrections to the EM27/SUN pressures as they were deployed at a slightly different altitude than the 125HR tracking mirror at the TCCON site. In these cases, we used the Digiquartz sensor pressure standard that was measuring at the same level as the EM27/SUNs to calculate the difference in surface pressure and added an offset of +0.1 hPa at AFRC and +0.3 hPa at Lamont to the original pressure value.

**Specific comments**

**RC:** *L48: I am not fully aware of TCCON doing calibration of surface pressure measurements. If they do, please provide a citation or an explanation how this is done.*

**AR:** As part of the TCCON guidelines the primary investigators at each station sites calibrate the pressure sensors against a pressure standard. There is an additive offset added to the measured pressures before running the retrievals. The pressure calibration data are not included in a publication.

**RC:** *L93: The abbreviations used for the EM27/SUNs and the TCCON station seem to be chosen randomly (e.g. it is unclear to me why the Armstrong Flight Research Center is abbreviated with "df"). Furthermore, for a reader it is quite confusing which abbreviation is a TCCON station and which is an EM27/SUN. Maybe add TCCON-xx to the TCCON sites or vice versa.*

**AR:** We recognise that there can be misunderstandings if the reader is unfamiliar with the two-letter TCCON station ID naming convention. We carefully reviewed the manuscript and made sure we clearly point out explicitly if we are referring to an EM27/SUN or a TCCON site.

**RC:** *Table 1: It would help to reduce the reader's confusion with the abbreviations if you would add the abbreviations of the TCCON sites in the "Site" column.*

AR:  We made the changes in Table 1 as suggested:

Table 1: This table lists the TCCON site locations and the dates the EM27/SUNs were on-site, the number of days with successful measurements, the average number of spectra collected by each EM27/SUN, and the total number of spectra collected by the TCCON instrument during the visit. The ILS column indicates whether EM27/SUN ILS measurements were collected at that location. The number of AirCore launches performed near the TCCON station during the dates listed is included in the final column. *instrument in brackets was not operational

| Site | Latitude °N | Longitude °W | Elevation (masl) | Dates | Days | EM27/SUNs* | Spectra count EM27/SUN | Spectra count TCCON | ILS | AirCore Launches |
|---|---|---|---|---|---|---|---|---|---|---|
| Caltech (ci) | 34.136 | 118.127 | 237 | 2018-07-06 – 2018-07-12 | 7 | ta,tb,tc,dn | 21356 | 1268 | Yes | - |
| AFRC (df) | 34.960 | 117.881 | 699 | 2018-07-13 – 2018-07-19 | 7 | ta,tb,tc,dn | 22667 | 2177 | No | 6 |
| Lamont (oc) | 36.605 | 97.486 | 320 | 2018-07-21 – 2018-07-29 | 5 | ta,tb,tc,dn | 17744 | 921 | Yes | 9 |
| Park Falls (pa) | 45.945 | 90.273 | 442 | 2018-07-31 – 2018-08-07 | 4 | (ta),tb,tc | 4436 | 406 | No | 4 |
| East Trout Lake (et) | 54.354 | 104.987 | 517 | 2018-08-09 – 2018-08-18 | 6 | ta,tb,tc | 14910 | 770 | Yes | - |
| Eureka (eu) | 80.053 | 86.417 | 610 | 2020-03-04 – 2020-08-31 | 61 | tb | 131713 | 5166 | No | - |

RC:  *L169 -– L171: I am not sure if I understand correctly what you are doing. This is what I understood: You are using the Digiquartz data as a „standard" measuring at height a. For AFRC and Lamont you are adding a factor to bring the pressure of the TCCON station to the level of the "standard" to correct for height. Have you ever compared the pressure measurements of the "standard" with the pressure of the TCCON station? Because otherwise, you cannot be sure if the correction is only due to height or also compensating sensor biases.*

AR:  We are assuming TCCON pressure sensors are already calibrated. We also tend not to make any changes to TCCON data that is used by a lot of users. Digiquartz is used to ensure the consistency which was observed at all sites except the differences due to the differences in height of measurements which was consistent with theoretical calculation.

RC:  *L290: In the list of citations Alberti et al. 2022 would be good do mention, too.*

AR:  We made the changes in Table 1 as suggested:

> Small biases in Xgas are expected to exist between the EM27/SUNs due to the differences in the instrument alignment (i.e., ILS) (Alberti et al., 2022). As long as these biases remain constant in time, including after shipping, a simple additive correction can place all the EM27/SUNs onto the same scale .

RC:  *Caption Figure 6: I was confused by this plot first, since I thought you are comparing something with the TCCON data and not that you only recorded the measurements at these sides. Maybe it is worth to write this explicitly.*

AR:  To make it more clear we made the change to the x-axis in figure 6 to represent different measurement locations instead of TCCON abbreviations to avoid further confusions. We have also updated the caption as follows:

> These figures show the median differences between the 10-minute averaged EM27/SUN $XCO_2$ (in ppm), $XCH_4$ and XCO (both in ppb) from coincident EM27/SUN measurements collected throughout the

2018 field campaign at each TCCON site before applying the instrument-to-instrument bias correction. The colour bars show the median biases relative to the "tb" EM27/SUN instrument at each stop during the field trip. Top panel shows GGG2014 and the bottom panel GGG2020 results respectively. The error bars on each colour bar represent the median absolute deviation in the 10-minute averaged biases.

**RC:** *Figure 8: Adding vertical lines separating the measurements of the different sites could help to make the plot clearer.*

AR: We have added vertical lines to the plot:

[Figure]

**RC:** *L365: Add explicitly which is the reference EM27/SUN (is it tb?). This could help the reader for making fast comparisons with the figures.*

AR: We made the changes in the text as suggested:

Figure 9 presents the median bias between the 10-minute-average Xgas values from the reference EM27/SUN ("tb") and each TCCON instrument.

**RC:** *L408: This sentence is more appropriate to section 4, not 4.1. Because it sums up the results of all the other stations than eureka, it appears misplaced to me in a section treating the peculiarities of the Eureka station.*

AR: We have moved the discussion about Eureka to section 4.1.

---

## Author Comment (AC3)

**Authors' Response to Reviews of**

**Using portable low-resolution spectrometers to evaluate TC-CON biases in North America**

Nasrin Mostafavi Pak, Jacob Hedelius, Sébastien Roche, Liz Cunningham, Bianca Baier, Colm Sweeney, Coleen Roehl, Joshua Laughner, Geoffrey Toon, Paul Wennberg, Harrison Parker, Colin Arrowsmith, Joseph Mendonca, Pierre Fogal, Tyler Wizenberg, Beatriz Herrera , Kimberly Strong, Kaley A. Walker, Paul Wennberg, Felix Vogel, Debra Wunch
*Atmospheric Measurement Techniques,*
* * *
**RC:** *Reviewers' Comment*,    AR: Authors' Response,    ☐ Manuscript Text

We would like to thank the reviewer for their thoughtful comments that have helped to improved our manuscript substantially. The detailed response and corrections are given below.

**Reviewer #2**

**General Comments**

**2.3 ILS measurements:**

RC: *Although ILS measurements using open path H2O spectra were made where possible, they are not used in any way in the retrievals. They seem to be used only qualitatively as an indication of any possible change in alignment. The version of linefit used (14.0, with full 20-parameter model) is different from that recommended by Alberti et al. (2021) (14.8, simple 2-parameter fit). The ME at maximum OPD derived here is not as unique as a measure of ILS as the simple 2 parameter version because it depends on the shape of the ME or phase vs OPD curves and on the SNR, regularisation and a priori assumptions. Thus the only real message from this section is that there was no major change of alignment from the 4 instruments (Fig 2, lower), and this section could possibly be shortened to reflect that.*

AR: Agreed. In this study ILS is only used qualitatively to ensure the optical alignment of the instruments haven't been changed significantly during the road trip after being transported from one site to the other. We have added additional explanation in this section for clarification:

> Instrumental Line Shape (ILS) is a measure of the optical alignment of the instrument and imperfections in this alignment can cause biases in the retrievals. The ILS of an FTS can be described by two parameters: Phase Error (PE) and Modulation Efficiency (ME). For high resolution FTS instruments (TCCON) ILS is typically reported as function of optical path difference (OPD) and for the low resolution EM27/SUNs these are typically reported at the maximum optical path difference. ILS values are not implemented into the GGG2014 and GGG2020 retrieval algorithms and are only used to evaluate the instruments' alignment qualitatively. TCCON guidelines require that the modulation efficiency deviates less than 5% from 1.0, over 0 to 45 cm OPD. A modulation loss of 1% in the EM27/SUN causes a bias of 0.1% in $XCO_2$ and 0.15% in $XCH_4$ (Hedelius et al., 2016), therefore we aim to ensure that the EM27/SUN ME variations remain less than 1%.

**3.1.1. Airmass dependent correction factors (ADCFs):**

**RC:** *Although the ADCF method is referenced, I think the formula/algorithm for the correction should be summarised here; there is otherwise no information in the paper about how the factors listed in Table 3 are applied and the reader has no idea how to interpret them. TCCON ADCFs also include a reference SZA around which the correction is expanded, and an exponent in the formula. It should be stated if/that these are fixed for EM27 analyses, and to which values. In line 267, perhaps reword slightly to make clearer that the derived ADCFs are obtained by fitting the long term 2018-2021 records from the 4 instruments to mean values for this dataset, and that these mean values are then applied to all further measurements.*

AR: We have added additional explanations and formula to make this section more clear:

> In this model, an asymmetric function ($A(t_i)$) representing true variations in Xgas over the course of the day and a symmetric function ($S(\theta_i)$) representing the airmass dependant artefact are used to fit to the daily measured Xgas values during the course of the day (Wunch et al. , 2011).
>
> $$y_i = \hat{y}[1 + \alpha A(t_i) + \beta S(\theta_i)] \tag{1}$$
>
> where $y_i$ is the Xgas from each spectrum and $\hat{y}$ is the mean value of $XCO_2$ on that day. $A(t_i)$ and $S(\theta_i)$ are defined as (Wunch et al. , 2011):
>
> $$A(t_i) = sin(2\pi(t_i - t_{noon})) \tag{2}$$
>
> where $t_i$ and $t_{noon}$ are in unit of days.
>
> $$S(\theta_i) = \left(\frac{\theta_i + \theta_0}{90 + \theta_0}\right)^p - \left(\frac{45 + \theta_0}{90 + \theta_0}\right)^p \tag{3}$$
>
> where $\theta_i$ is in degrees and $\theta_0$ and $p$ are empirically found to be $13°$ and 3, respectively. $\alpha$ and $\beta$ are found by minimizing the difference between the measured $y_i$ and the fitted functions. Airmass dependent correction is then applied to Xgas by:
>
> $$y_c = \frac{y_i}{[1 + \beta S(\theta_i)]} \tag{4}$$
>
> where $\beta$ is the airmass dependant correction factor (ADCF) and $y_c$ is the airmass corrected Xgas value.

**3.1.2 Instrument-instrument biases and "calibration":**

**RC:** *Are the bias corrections to the raw measurements applied multiplicatively or additively? As with the ADCF, it would be helpful to spell out near line 300 the algebraic algorithm used for this (and every) correction. In the COCCON network a similar process is followed, and though Frey 2019 is referenced, it is not discussed or compared. I think it would be helpful to add some text to compare the approach here with that of COCCON, for example to (hopefully) demonstrate that both groups see similar biases. Ultimately the*

*research community that uses these data needs to know that biases between instruments not only within the N American and European-centred networks are minimised, but also that they also minimised between the networks. The two networks use different retrieval codes, so it is quite important to be confident they do not introduce any relative bias.*

AR: In this study the bias correction is applied additively. We have briefly summarized the Frey et al. results for comparison:

> The range of the biases are up to 0.2 ppm in $XCO_2$, 4 ppb for $XCH_4$ and 0.8 ppb for XCO. In a study by Frey et al. (2019b), biases between 30 EM27/SUNs were evaluated and a scale factors of 0.999-1.0004 for XCO2 and 0.9975-1.0026 for $XCH_4$ were found. This would be equivalent to an average bias of 0.6 ppm for $XCO_2$ for an average DMF of 400 ppm and 9 ppb for $XCH_4$ for an average DMF of 1840 ppb. We observe smaller biases between our instruments, which might be expected as we only compare 4 EM27/SUNs.

**4. Evaluation of TCCON biases against the EM27/SUNs (and Discussion):**

RC: *This section could be improved and more readable by reorganising to separate out the "comparison with TCCON" from the "discussion" of the interpretation of the time series such as at Eureka. For example the paragraph from L 348 discusses the atmospheric interpretation of the Eureka time series, then the discussion returns to the TCCON-EM27 biases in Fig 10 and Table 6. Perhaps this paragraph could be merged into the current section 4.1? In the TCCON-EM27 bias section, since Xluft has been added to Fig 8, I would recommend also adding the mean Xluft values to Table 5 so the reader can do their own sanity check to see just how different from 1.000 they are.*

AR: As also suggested by reviewer 1, we have moved the discussion about Eureka to section 4.1. We also have added a table to present the average xluft values:

Table 1: Average Xluft values measured by TCCON (125 HR) and the reference EM27/SUN (tb)

| TCCON Site | Average TCCON Xluft | Average EM27/SUN Xluft (tb) |
|---|---|---|
| Caltech (ci) | 0.9993 | 1.0020 |
| AFRC (df) | 0.9979 | 1.0014 |
| Lamont (oc) | 0.9990 | 1.0022 |
| Park Falls (pa) | 1.0028 | 1.0013 |
| East Trout Lake (et) | 1.0002 | 1.0040 |
| Eureka (eu) | 1.0005 | 1.0066 |

**5. Conclusions and general:**

RC: *I would find it helpful to make more explicit comparisons with similar studies on the COCCON network, for example by Frey et al 2019, to the extent allowed by the COCCON publications. At present COCCON is acknowledged but not further compared or discussed. I am sure the modelling community would like to be able to combine results from both this work (using GGG and EGI for analysis) and COCCON (using*

*Proffast and independent post-processing) without fear of significant bias. Explicitly addressing this point of comparison would be particularly useful.*

AR: We have briefly summarized the Frey et al. results for comparisons.

> Frey et al. (2019b) performed comparisons between an EM27/SUN at Karlsruhe Institute of Technology (KIT) and the KIT TCCON instrument and found a scaling factor of 1.0098 for $XCO_2$ for high resolution TCCON and 1.0014 for low resolution TCCON retrieval equivalent to 4 ppm and 0.6 ppm respectively. For $XCH_4$ a scaling factor of 1.0072 was found for high resolution TCCON and 0.9997 for low resolution TCCON equivalent to 13 ppb and 0.5 ppb respectively. One should note that the high resolution TCCON spectra were processed using GGG2014, whereas the low resolution TCCON and the EM27/SUN retrievals were performed using a different retrieval algorithm (PROFFIT version 9.6), which could explain the larger bias found between the EM27/SUN and TCCON in the Frey et al. work, compared to this study.

**Specific comments**

RC: *L45: "First and foremost..." Actually, TCCON stations "first and foremost" use consistent spectrometer hardware and data collection protocols to collect spectra, before using consistent data analysis methods.*

AR: We have made changes in text as recommended:

> There are certain practices in place to ensure site-to-site consistency between TCCON observations. First and foremost, each TCCON station is equipped with nearly identical spectrometer hardware, and each dataset is analyzed using a consistent version of the GGG software, including identical spectroscopy.

RC: *L54: cm-1 not italicised*

AR: We have made changes in text as recommended throughout the text:

> The EM27/SUNs (by Bruker Optics GmbH) are portable solar-viewing FTS instruments with a lower spectral resolution (0.5 $cm^{-1}$) than TCCON (0.02 $cm^{-1}$) that can be used to measure total column abundances of $CO_2$, $CH_4$, $H_2O$, and CO in...

RC: *L82: The original design includes (not consists of) one InGaAs detector...*

AR: We have made changes in text as recommended:

> The original design includes one InGaAs detector, and records in the spectral range of ...

RC: *L104: use "there is" rather than the abbreviation "there's" in formal text such as here.*

AR: We have made changes in text as recommended:

> TCCON instruments run automatically during cloud-free times of the day when there is sufficient sunlight.

**RC:** *L106: Please state here, maybe better in section 2.2 or later site-by-site that the EM27/SUNs were at the same height asl as the TCCON trackers, or if not, how the altitude-dependent pressure correction was applied.*

AR: In section 2.2, we have clarified how we used the standard pressure sensor to account for differences in pressure due to differences in height:

> For the EM27/SUN retrievals, we use the pressure measurements recorded by the local TCCON weather stations which have previously been calibrated against a pressure standard, with the exception of Park Falls where we used the Vaisala WXT536 pressure data for both TCCON and EM27/SUNs, since the pressure measurements made by the TCCON pressure sensor were not stable during the campaign. In addition, at AFRC and Lamont we applied additional corrections to the EM27/SUN pressures as they were deployed at a slightly different altitude than the 125HR tracking mirror at the TCCON site. In these cases, we used a Digiquartz sensor pressure standard that was measuring at the same altitude as the EM27/SUNs to calculate the difference in surface pressure and added an offset of +0.1 hPa at AFRC and +0.3 hPa at Lamont to the original pressure value.

**RC:** *L157: see L104 - it is*

AR: We have made changes in text as recommended:

> Therefore, it is necessary to ensure that the pressure measurements are accurate by calibrating the local TCCON pressure sensors ...

**RC:** *L197: replace "high accuracy" with "traceable accuracy"*

AR: We have made changes in text as recommended:

> In order to achieve traceable accuracy, total column measurements ....

**RC:** *L198: add "calibrated" and "simultaneously" : . . . against calibrated airborne in situ profiles that are simultaneously collected at the TCCON sites.*

AR: We have made changes in text as recommended:

> ... total column measurements from TCCON are tied to the WMO trace gas scale by comparing against calibrated airborne in situ measurement profiles that are simultaneously collected at the TCCON sites ...

**RC:** *L201: update ref to Tans 2009 – there was a recent and complete paper by Tans describing the theory behind Aircore.*

AR: We have made changes in text as recommended:

> An alternative method for obtaining vertical profiles that extend higher is to use the AirCore sampling system (Tans, 2009; Karion et al., 2010; Tans, 2022).

**RC:** *L203: I think the aircore description is a little too brief - it is worth spelling out here that the aircore tube fills from one end so the earliest sample is compressed into the far end of the tube, and that diffusion is slow enough that the vertical profile is preserved along the tube length. "What about diffusion?" is the most common question I get when explaining Aircore to people.*

AR: We have added some more information about the AirCore to clarify the reviewer's point:

> In this method, a coiled 100-meter long hollow tube with a small inner diameter of about 0.2–0.3 cm is launched using a balloon. The AirCore is filled with a mixture of known trace gas mole fractions of interest prior to launch, and this gas evacuates during ascent. Upon descent, the nearly-empty AirCore fills with ambient air where the earliest sample is compressed into the far end of the tube. Because molecular diffusion and Taylor dispersion acts slowly within this tubing coil (Tans, 2022), there is little mixing of the continuous air sample collected within the tube and the vertical profile of the atmosphere is preserved within the tube. The tube is then sealed upon landing, retrieved and quickly analyzed using a Picarro Cavity Ring-Down Spectrometer. AirCore altitude ceilings for balloon flights are typically set to 30 km asl, with trace gas profiles derived from approximately 27 km to the surface (Karion et al., 2010).

**RC:** *L216: launches is presented . . . (not are)*

AR: We have made changes in text as recommended:

> A summary of dates, times and locations of the AirCore launches is presented in ...

**RC:** *L220 and L 225: why do you use the a priori GGG profile above the aircore ceiling rather than the scaled a priori profile after the fit? How significant is the difference? Similarly in Figure 3, why show the a priori profile and not the scaled profile after the fit?*

AR: We assume that atmospheric variability in $XCO_2$ and $XCH_4$ is largest near surface and smallest in the upper stratosphere and mesosphere, and therefore the scaling retrieval is more responsive to changes nearer to the surface than above the AirCore ceiling. We further assume that the prior aloft is reasonably close to truth, so we use the original prior to extend the AirCore profile up to 70 km.

The aim of Figure 3 is to show an example of the shape of the standard GGG prior and compare it with the true profile measured by the AirCore. We assume that the scaling factor retrieved when using the standard a priori includes contributions from spectroscopy errors and from real difference between the a priori profile and the true atmospheric profile. If we replace the standard a priori profile with the AirCore profile, we assume that the retrieved scaling factor represents only the scaling caused by spectroscopy errors. Therefore, to compute the AICF, we rerun gfit using the AirCore profile as the a priori to compute the AICF values. In Figure 3, we present examples of the profiles that are used as a priori profiles in GGG.

**RC:** *L231: perhaps add ". . . spectroscopic linelist. . . " Linelist may be jargon to some readers.*

AR: We have made changes in text as recommended:

> ..., and changes to the spectroscopy include improvements in the spectroscopic linelist, ...

**RC:** *L241: spell out GEOS-FPIT on first usage*

AR: We have made changes in text as recommended:

> For GGG2020, 3-hourly vertical profiles are obtained from Goddard Earth Observing System- Forward Processing for Instrument Teams (GEOS-FPIT) atmospheric data assimilation system ...

**RC:** *L262: suggest reword to "... caused by spectroscopic and instrumental inaccuracies." Or "errors".*

AR: We have made changes in text as recommended:

> Retrieved Xgas values have a small solar zenith angle (or airmass) dependence caused by spectroscopic and instrumental inaccuracies.

**RC:** *L278: replace "deteriorates" with "increases" or "actually increases"*

AR: We have made changes in text as recommended:

> ..., and applying the TCCON correction factors increases the airmass dependence.

**RC:** *L324: The aircore does not "float", this term comes from ballon-borne occultation measurements. Perhaps replace "float" with "maximum"*

AR: Changed to maximum

**RC:** *L332: can you provide a reference to justify why you can state that the altitude errors in the aircore data are negligible?*

AR: We integrated the AirCore after applying the altitude error and the numbers are based on our calculations. We have modified the text to make this point more clear:

> The errors associated with altitude error were calculated by shifting the AirCore profile upward and downward by using the altitude error associated with each gas at each level and ceiling altitude error was calculated by integrating the profile above the AirCore ceiling assuming an error of 1 km. The errors due to altitude error and ceiling altitude error are negligible with orders of magnitude smaller than $10^{-4}$ % for $CO_2$, $10^{-3}$ % for $CH_4$ and 0.01 % for CO.

**RC:** *L342: In the definition of Xluft, please indicate that Vdryair is calculated from measured surface pressure at the time of the measurement.*

AR: We have made changes in text as recommended:

> Xluft is defined as $Xluft = 0.2095 \times V_{\text{dry air}}/V_{O_2}$ where $V$ indicates a column density (in molecules per cm$^2$). $V_{\text{dry air}}$ is calculated from measured surface pressure at the time of measurement.

**RC:** *Caption Figure 8 and 9. To be clear, I suggest you add "collocated" as in "Timeseries of EM27/SUN (tb) retrieved XCO2, XCH4, XCO and Xluft in color and co-located TCCON in...".*

AR: We have made changes in text as recommended:

Figure 8. Timeseries of EM27/SUN (tb) retrieved $XCO_2$, $XCH_4$, XCO and Xluft in colour and co-located TCCON in transparent grey during the summer 2018 campaign (GGG2020). Different sites are highlighted by a different colour.

Figure 9. Timeseries of EM27/SUN (tb) and co-located TCCON retrieved $XCO_2$, $XCH_4$, XCO, Xluft and XHF (TCCON only) during the spring and summer 2020 Eureka campaign (GGG2020).

**RC:** *L357: the meaning of this statement is not clear to me – reduced by a factor of 8 (2) relative to what? Single spectrum measurements, I assume – please state so. This is not so important, it is the averaging time that matters most.*

AR: We have added more explanation to clarify this point:

Assuming measurements are dominated by Gaussian noise, variability in the EM27/SUN averages will be reduced by a factor of 8, and variability in the TCCON averages will be reduced by a factor of 2 relative to single spectrum measurements.

**RC:** *L438: When using GGG2020, state explicitly if the site to site biases agree everywhere for both high and low resolution TCCON measurements. (From Table 6, the maximum biases are 0.53 and 0.8 ppm respectively – these are not strictly less than the variability quoted of 0.5 ppm)*

AR: We have made changes in text as recommended:

When using GGG2020, the site-to-site biases in the high resolution TCCON agree everywhere within the expected variability.